# Private Everlasting Prediction

Moni Naor[*]        Kobbi Nissim[†]        Uri Stemmer[‡]        Chao Yan[§]

## Abstract

A private learner is trained on a sample of labeled points and generates a hypothesis that can be used for predicting the labels of newly sampled points while protecting the privacy of the training set [Kasiviswannathan et al., FOCS 2008]. Past research uncovered that private learners may need to exhibit significantly higher sample complexity than non-private learners as is the case of learning of one-dimensional threshold functions [Bun et al., FOCS 2015, Alon et al., STOC 2019].

We explore prediction as an alternative to learning. A predictor answers a stream of classification queries instead of outputting a hypothesis. Earlier work has considered a private prediction model with a single classification query [Dwork and Feldman, COLT 2018]. We observe that when answering a stream of queries, a predictor must modify the hypothesis it uses over time, and in a manner that cannot rely solely on the training set.

We introduce *private everlasting prediction* taking into account the privacy of both the training set *and* the (adaptively chosen) queries made to the predictor. We then present a generic construction of private everlasting predictors in the PAC model. The sample complexity of the initial training sample in our construction is quadratic (up to polylog factors) in the VC dimension of the concept class. Our construction allows prediction for all concept classes with finite VC dimension, and in particular threshold functions over infinite domains, for which (traditional) private learning is known to be impossible.

## 1 Introduction

A PAC learner for a concept class $C$ is given labeled examples $S = \{(x_i, y_i)\}_{i \in [n]}$ drawn i.i.d. from an unknown underlying probability distribution $\mathcal{D}$ over a data domain $X$ and outputs a hypothesis $h$ that can be used for predicting the label of fresh points $x_{n+1}, x_{n+2}, \ldots$ sampled from the same underlying probability distribution $\mathcal{D}$ [Valiant, 1984]. It is well known that when points are labeled by a concept selected from a concept class $C = \{c : X \to \{0, 1\}\}$ then learning is possible with sample complexity proportional to VC($C$).

[*]Department of Computer Science and Applied Math, Weizmann Institute of Science. `moni.naor@weizmann.ac.il`. Incumbent of the Judith Kleeman Professorial Chair. Research supported in part by grants from the Israel Science Foundation (no.2686/20), by the Simons Foundation Collaboration on the Theory of Algorithmic Fairness and by the Israeli Council for Higher Education (CHE) via the Weizmann Data Science Research Center.

[†]Department of Computer Science, Georgetown University. `kobbi.nissim@georgetown.edu`. Research supported in part by NSF grant No. CNS-2001041 and a gift to Georgetown University.

[‡]Blavatnik School of Computer Science, Tel Aviv University, and Google Research. `u@uri.co.il`. Partially supported by the Israel Science Foundation (grant 1871/19) and by Len Blavatnik and the Blavatnik Family foundation.

[§]Department of Computer Science, Georgetown University. `cy399@georgetown.edu`. Research supported in part by by a gift to Georgetown University.

Learning often happens in settings where the underlying training data is related to individuals and privacy-sensitive. For legal, ethical, or other reasons, the learner is then required, to protect personal information from being leaked in the learned hypothesis $h$. Private learning was introduced by Kasiviswanathan et al. [2011] as a theoretical model for studying such tasks. A *private learner* is a PAC learner that preserves differential privacy with respect to its training set $S$. That is, the learner's distribution on outcome hypotheses must not depend too strongly on any single example in $S$. Kasiviswanathan et al. showed that any finite concept class can be learned privately and with sample complexity $n = O(\log|C|)$, a value that is sometimes significantly higher than the VC dimension of the concept class $C$.

It is now understood that the gap between the sample complexity of private and non-private learners is essential – an important example is private learning of threshold functions (defined over an ordered domain $X$ as $C_{thresh} = \{c_t\}_{t \in X}$ where $c_t(x) = \mathbb{1}_{x \geq t}$), which requires sample complexity that is asymptotically higher than the (constant) VC dimension of $C_{thresh}$. In more detail, with *pure* differential privacy, the sample complexity of private learning is characterized by the representation dimension of the concept class [Beimel et al., 2013a]. The representation dimension of $C_{thresh}$ (hence, the sample complexity of private learning thresholds) is $\Theta(\log|X|)$ [Feldman and Xiao, 2015]. With *approximate* differential privacy, the sample complexity of learning threshold functions is $\Theta(\log^*|X|)$ [Beimel et al., 2013b, Bun et al., 2015, Alon et al., 2019, Kaplan et al., 2020, Cohen et al., 2022]. Hence, whether with pure or with approximate differential privacy, the sample complexity of privately learning thresholds grows with the cardinality of the domain $|X|$ and no private learner exists for this task over infinite domains. In contrast, non-private learning is possible with constant sample complexity (independent of $|X|$).

**Privacy preserving (black-box) prediction.** Dwork and Feldman [2018] proposed privacy-preserving prediction as an alternative for private learning. Noting that "[i]t is now known that for some basic learning problems [. . .] producing an accurate private model requires much more data than learning without privacy," they considered a setting where "users may be allowed to query the prediction model on their inputs only through an appropriate interface". That is, a setting where the learned hypothesis is not made public and may be accessed only in a *black-box* manner via a privacy-preserving query-answering prediction interface. The prediction interface is required to preserve the privacy of its training set $S$:

**Definition 1.1** (private prediction interface [Dwork and Feldman, 2018] (rephrased)). A prediction interface $\mathcal{A}$ is $(\epsilon, \delta)$-differentially private if for every interactive query generating algorithm $Q$, the output of the interaction between $Q$ and $\mathcal{A}(S)$ is $(\epsilon, \delta)$-differentially private with respect to $S$.

Dwork and Feldman focused on the setting where the entire interaction between $Q$ and $\mathcal{A}(S)$ consists of issuing a single prediction query and answering it:

**Definition 1.2** (Single query prediction [Dwork and Feldman, 2018]). Let $\mathcal{A}$ be an algorithm that given a set of labeled examples $S$ and an unlabeled point $x$ produces a label $y$. $\mathcal{A}$ is an $(\epsilon, \delta)$-differentially private prediction algorithm if for every $x$, the output $\mathcal{A}(S, x)$ is $(\epsilon, \delta)$-differentially private with respect to $S$.

W.r.t. answering a single prediction query, Dwork and Feldman showed that the sample complexity of such predictors is proportional to the VC dimension of the concept class.

## 1.1 Our contributions

We extend private prediction to answering a sequence – *unlimited in length* – of prediction queries. We refer to this as *private everlasting prediction* (PEP). Our goal is to present a generic private everlasting predictor with low training sample complexity $|S|$.

### 1.1.1 Private prediction interfaces when applied to a large number of queries

We begin by examining private everlasting prediction under the framework of Definition 1.1. We prove:

**Theorem 1.3** (informal version of Theorem 3.3). *Let $\mathcal{A}$ be a private everlasting prediction interface for concept class $C$ and assume $\mathcal{A}$ bases its predictions solely on the initial training set $S$, then there exists a private learner for concept class $C$ with sample complexity $|S|$.*

This means that everlasting predictors that base their prediction solely on the initial training set $S$ are subject to the same complexity lowerbounds as private learners. Hence, to avoid private learning lowerbounds, private everlasting predictors need to rely on more than the initial training sample $S$ as a source of information about the underlying probability distribution and the labeling concept.

In this work, we choose to allow the everlasting predictor to rely on the queries made – which are unlabeled points from the domain $X$ – assuming the queries are drawn from the same distribution the initial training $S$ is sampled from. This requires modifying the privacy definition, as Definition 1.1 does not protect the queries issued to the predictor.

### 1.1.2 A definition of private everlasting predictors

Our definition of private everlasting predictors is motivated by the observations above. Consider an algorithm $\mathcal{A}$ that is first fed with a training set $S$ of labeled points and then executes for an unlimited number of rounds, where in round $i$ algorithm $\mathcal{A}$ receives as input a query point $x_i$ and produces a label $\hat{y}_i$. We say that $\mathcal{A}$ is an everlasting predictor if, when the (labeled) training set $S$ and the (unlabeled) query points are coming from the same underlying distribution, $\mathcal{A}$ answers each query points $x_i$ by invoking a good hypothesis $h_i$ (i.e., $h_i$ has low generalization error), and hence the label $\hat{y}_i$ produced by $\mathcal{A}$ is correct with high probability. We say that $\mathcal{A}$ is a *private* everlasting predictor if its sequence of predictions $\hat{y}_1, \hat{y}_2, \hat{y}_3, \ldots$ protects both the privacy of the training set $S$ *and* the query points $x_1, x_2, x_3, \ldots$ in face of any adversary that chooses the query points adaptively.

We emphasize that while private everlasting predictors need to exhibit average-case utility – as good prediction is required only for the case where the initial training set $S$ and the queries $x_1, x_2, x_3, \ldots$ are selected i.i.d. from the same underlying distribution – our privacy requirement is worst-case, and holds in face of an adaptive adversary that chooses each query point $x_i$ after receiving the prediction provided for $(x_1, \ldots, x_{i-1})$, and not necessarily in accordance with any probability distribution.

### 1.1.3 A generic construction of private everlasting predictors

We show a reduction from private everlasting prediction of a concept class $C$ to non-private PAC learning of $C$. Our Algorithm `GenericBBL`, presented in Section 6, executes in rounds.

**Initialization.** The input to the first round is a labeled training set $S$ where $|S| = O\big((\text{VC}(C))^2\big)$, assumed to be labeled consistently with some unknown target concept $c \in C$. Denote $S_1 = S$ and $c_1 = c$.

**Rounds.** Each round begins with a collection $S_i$ of labeled examples and ends with a newly generated collection of labeled examples $S_{i+1}$ that feeds as input for the next round. The size of these collections grows by a constant factor at each round, to allow for the accumulated error of the predictor to converge. The construction ensures that each labeled set $S_i$ is consistent with some concept in $c_i \in C$, which is not necessarily the original target concept $c$, but has a bounded generalization error with respect to $c$. In more detail, the construction ensures that $\text{Pr}_{x \sim \mathcal{D}}[c_{i+1}(x) \neq c_i(x)]$ decreases by a factor of two in every round, and hence, by the triangle inequality, $\text{Pr}_{x \sim \mathcal{D}}[c_i(x) \neq c(x)]$ is bounded.

We briefly describe the main computations performed in each round of `GenericBBL`.[5]

- **Round initialization:** At the outset of a round, the labeled set $S_i$ is partitioned into sub-sets, each with number of samples which is proportional to the VC dimension (so we have $\approx \frac{|S_i|}{\text{VC}(C)}$ sub-sets). Each of the sub-sets is used for training a classifier

---

[5]Important details, such as privacy amplification via sampling and management of the learning accuracy and error parameters are omitted from the description provided in this section.

non-privately, hence creating a collection of classifiers $F_i = \{f : X \to \{0, 1\}\}$ that are used throughout the round.

- **Query answering:** Queries are issued to the predictor in an online manner. A query is first labeled by each of the classifiers in $F_i$. Then the predicted label is computed by applying a privacy-preserving majority vote on these intermediate labels. By standard composition theorems for differential privacy, we could answer roughly $|F_i|^2 \approx \left(\frac{|S_i|}{\mathrm{VC}(C)}\right)^2$ queries without exhausting the privacy budget.

- **Generating a labeled set for the following round:** A round ends with the preparation of a collection $S_{i+1}$ of labeled samples that is to be used in the initialization of next round. We explore alternatives for creating $S_{i+1}$:

  - As at the end of a round the predictor has already labeled all the queries presented during the round's execution, one alternative is to let $S_{i+1}$ consist of these queries and the labels provided for them. Note, however, that it is not guaranteed that the majority vote would result in a set of labels that are consistent with some concept in $C$, even if $S_i$ is consistent with some concept in $C$. Hence, following this alternative would require the use of non-private *agnostic* learners instead. Furthermore, the error introduced by the majority vote can be larger than the generalization error of the classifiers in $F_i$ by a constant factor greater than one. This may prevent the generalization error of the classifiers used in future rounds from converging.[6]

  - To overcome this problem, we use Algorithm `LabelBoost` – a tool developed by Beimel et al. [2021] in the context of private semi-supervised learning. `LabelBoost` takes as input the sample $S_i$ (labeled by a concept $c_i \in C$) and the (unlabeled) queries made during the round. It labels them with a concept $c_{i+1} \in C$ where the error of $c_{i+1}$ with respect to $c_i$, i.e., $\Pr_{x \sim \mathcal{D}}[c_{i+1}(x) \neq c_i(x)]$ is bounded.

- **Controlling the generalization error:** Let $S_{i+1}$ denote the (re)labeled query points obtained in the $i$th round. This is a collection of size $|S_{i+1}| \approx \left(\frac{|S_i|}{\mathrm{VC}(C)}\right)^2$. Hence, provided that $|S_i| \gtrsim (\mathrm{VC}(C))^2$ we get that $|S_{i+1}| > |S_i|$. This allows to lower the accuracy parameters of the non-private learners in each round, and hence ensure that the total error converges.

**Theorem 1.4** (informal version of Theorem 6.1). *For every concept class $C$, Algorithm* `GenericBBL` *is a private everlasting predictor requiring an initial set of labeled examples which is (upto polylogarithmic factors) quadratic in the VC dimension of $C$.*

## 1.2 Related work

Beyond the work of Dwork and Feldman [2018] on private prediction mentioned above, our work is related to private semi-supervised learning and joint differential privacy.

**Semi-supervised private learning.** As in the model of private semi-supervised learning of Beimel et al. [2021], our predictors depend on both labeled and unlabeled samples. Beyond the difference between outputting a hypothesis and providing black-box prediction, a major difference between the settings is that in the work of Beimel et al. [2021] all samples – labeled and unlabeled – are given at once at the outset of the learning process whereas in the setting of everlasting predictors the unlabeled samples are supplied in an online manner. Our construction of private everlasting predictors uses tools developed for the semi-supervised setting, and in particular Algorithm `LabelBoost` of of Beimel et al.

**Joint differential privacy.** Kearns et al. [2015] introduced joint differential privacy (JDP) as a relaxation of differential privacy applicable for mechanism design and games. For

---

[6] to see that the majority vote can increase the generalization error by a constant factor consider three classifiers $f_1, f_2, f_3$, each with generalization error $\alpha$, and let $A, B, C$ be three disjoint subsets of the support of the underlying distribution, each of probability weight $\alpha/2$. If the classifier $f_1$ errs on inputs in $A \cup B$, the classifier $f_2$ errs on $A \cup C$, and $f_3$ errs on $B \cup C$ then the majority of $f_1, f_2, f_3$ errs on inputs in $A \cup B \cup C$ and hence the generalization error grows from $\alpha$ to $1.5\alpha$.

every user $u$, JDP requires that the outputs jointly seen by all *other* users would preserve differential privacy w.r.t. the input of $u$. Crucially, in JDP users select their inputs ahead of the computation. In our settings, the inputs to a private everlasting predictor are prediction queries which are chosen in an online manner, and hence a query can depend on previous queries and their answers. Yet, similarly to JDP, the outputs provided to queries not performed by a user $u$ should jointly preserve differential privacy w.r.t. the query made by $u$. Our privacy requirement hence extends JDP to an adaptive online setting.

**Additional works on private prediction.** Bassily et al. [2018] studied a variant of the private prediction problem where the algorithm takes a labeled sample $S$ and is then required to answer $m$ prediction queries (i.e., label a sequence of $m$ unlabeled points sampled from the same underlying distribution). They presented algorithms for this task with sample complexity $|S| \gtrsim \sqrt{m}$. This should be contrasted with our model and results, where the sample complexity is independent of $m$. The bounds presented by Dwork and Feldman [2018] and Bassily et al. [2018] were improved by Dagan and Feldman [2020] and by Nandi and Bassily [2020] who presented algorithms with improved dependency on the accuracy parameter in the agnostic setting.

## 1.3 Discussion and open problems

We show how to transform any (non-private) learner for the class $C$ (with sample complexity proportional to the VC dimension of $C$) to a private everlasting predictor for $C$. Our construction is not polynomial time due to the use of Algorithm `LabelBoost`, and requires an initial set $S$ of labeled examples which is quadratic in the VC dimension. We leave open the question whether $|S|$ can be reduced to be linear in the VC dimension and whether the construction can be made polynomial time. A few remarks are in order:

1. While our generic construction is not computationally efficient, it does result in efficient learners for several interesting special cases. Specifically, algorithm `LabelBoost` can be implemented efficiently whenever given an input sample $S$ it is possible to efficiently enumerate all possible dichotomies from the target class $C$ over the points in $S$. In particular, this is the case for the class of 1-dim threshold functions $C_{thresh}$, as well as additional classes with constant VC dimension. Another notable example is the class $C_{thresh}^{enc}$ which intuitively is an "encrypted" version of $C_{thresh}$. Bun and Zhandry [2016] showed that (under plausible cryptographic assumptions) the class $C_{thresh}^{enc}$ cannot be learned privately and efficiently, while it can be learned efficiently non-privately. Our construction can be implemented efficiently for this class. This provides an example where private everlasting prediction can be done efficiently, while (standard) private learning is possible but necessarily inefficient.

2. It is now known that some learning tasks require the produced model to memorize parts of the training set in order to achieve good learning rates, which in particular disallows the learning algorithm from satisfying (standard) differential privacy [Brown et al., 2021]. Our notion of private everlasting prediction circumvents this issue, since the model is never publicly released and hence the fact that it must memorize parts of the sample is not of a direct privacy threat. In other words, our work puts forward a private learning model which, in principle, allows memorization. This could have additional applications in broader settings.

3. As mentioned above, in general, private everlasting predictors cannot base their predictions solely on the initial training set, and in this work we choose to rely on the *queries* presented to the algorithm (in addition to the training set). Our construction can be easily adapted to a setting where the content of the blackbox is updated based on *fresh unlabeled samples* (whose privacy would be preserved), instead of relying on the query points themselves. This might be beneficial to avoid poisoning attacks via the queries.

## 2 Preliminaries

### 2.1 Preliminaries from differential privacy

**Definition 2.1** (($\epsilon, \delta$)-indistinguishability)**.** Let $R_0, R_1$ be two random variables over the same support. We say that $R_0, R_1$ are ($\epsilon, \delta$)-indistinguishable if for every event $E$ defined over the support of $R_0, R_1$,

$$\Pr[R_0 \in E] \le e^\epsilon \cdot \Pr[R_1 \in E] + \delta \quad \text{and} \quad \Pr[R_1 \in E] \le e^\epsilon \cdot \Pr[R_0 \in E] + \delta.$$

**Definition 2.2.** Let $X$ be a data domain. Two datasets $x, x' \in X^n$ are called *neighboring* if $|\{i : x_i \ne x_i'\}| = 1$.

**Definition 2.3** (differential privacy [Dwork et al., 2006])**.** A mechanism $M : X^n \to Y$ is ($\epsilon, \delta$)-differentially private if $M(x)$ and $M(x')$ are ($\epsilon, \delta$)-indistinguishable for all neighboring $x, x' \in X^n$.

In our analysis, we use the post-processing and composition properties of differential privacy, that we cite in their simplest forms.

**Proposition 2.4** (post-processing)**.** *Let $M_1 : X^n \to Y$ be an ($\epsilon, \delta$)-differentially private algorithm and $M_2 : Y \to Z$ be any algorithm. Then the algorithm that on input $x \in X^n$ outputs $M_2(M_1(x))$ is ($\epsilon, \delta$)-differentially private.*

**Proposition 2.5** (composition)**.** *Let $M_1$ be a ($\epsilon_1, \delta_1$)-differentially private algorithm and let $M_2$ be ($\epsilon_2, \delta_2$)-differentially private algorithm. Then the algorithm that on input $x \in X^n$ outputs $(M_1(x), M_2(x)$ is ($\epsilon_1 + \epsilon_2, \delta_1 + \delta_2$)-differentially private.*

**Theorem 2.6** (Advanced composition Dwork et al. [2010b])**.** *Let $M_1, \ldots, M_k : X \to Y$ be ($\epsilon, \delta$)-differentially private algorithms. Then the algorithm that on input $x \in X$ outputs $(M_1(x), \ldots, M_k(x))$ is ($\epsilon', k\delta + \delta'$)-differentially private, where $\epsilon' = \sqrt{2k\ln(1/\delta')} \cdot \epsilon$ for every $\delta' > 0$.*

**Definition 2.7** (Laplace mechanism [Dwork et al., 2006])**.** For $f : X^n \to \mathbb{R}$ Let $\Delta_f = \max(f(x) - f(x'))$, where the maximum is taken over all neighboring $x, x' \in X^n$. The Laplace mechanis is $M(x) = f(x) + Y$, where $Y$ is sampled from the laplace distribution $\mathrm{Lap}(\Delta_f/\epsilon)$. The Laplace mechanism is ($\epsilon, 0$)-differentially private.

**Definition 2.8** (Exponential mechanism [McSherry and Talwar, 2007])**.** Let $q : X^n \times Y \to \mathbb{R}$ be a score function defined over data domain $X$ and output domain $Y$. Define $\Delta = \max(|q(x, y) - q(x', y)|)$ where the maximum is taken over all $y \in Y$ and all neighbouring databases $x, x' \in X^n$. The exponential mechanism is the ($\epsilon, 0$)-differentially private mechanism which selects an output $y \in Y$ with probability proportional to $e^{\frac{\epsilon q(x,y)}{2\Delta}}$.

**Claim 2.9** (Privacy amplification by sub-sampling [Kasiviswanathan et al., 2011])**.** *Let $\mathcal{A}$ be an ($\epsilon', \delta'$)-differentially private algorithm operating on a database of size $n$. Let $\epsilon \le 1$ and let $t = \frac{n}{\epsilon}(3 + exp(\epsilon'))$. Construct an algorithm $\mathcal{B}$ operating the database $D = (z_i)_{i=1}^t$. Algorithm $\mathcal{B}$ randomly selects a subset $J \subseteq \{1, 2, \ldots, t\}$ of size $n$, and executes $\mathcal{A}$ on $D_J = (z_i)_{i \in J}$. Then $\mathcal{B}$ is $\left(\epsilon, \frac{4\epsilon}{3 + exp(\epsilon')}\delta'\right)$-differentially private.*

### 2.2 Preliminaries from PAC learning

A concept class $C$ over data domain $X$ is a set of predicates $c : X \to \{0, 1\}$ (called concepts) which label points of the domain $X$ by either 0 or 1. A learner $\mathcal{A}$ for concept class $C$ is given $n$ examples sampled i.i.d. from an unknown probability distribution $\mathcal{D}$ over the data domain $X$ and labeled according to an unknown target concept $c \in C$. The learner should output a hypothesis $h : X \to [0, 1]$ that approximates $c$ for the distribution $\mathcal{D}$. More formally,

**Definition 2.10** (generalization error)**.** The *generalization error* of a hypothesis $h : X \to [0, 1]$ with respect to concept $c$ and distribution $\mathcal{D}$ is defined as $\mathrm{error}_{\mathcal{D}}(c, h) = \mathrm{Exp}_{x \sim \mathcal{D}}[|h(x) - c(x)|]$.

**Definition 2.11** (PAC learning [Valiant, 1984])**.** Let $C$ be a concept class over a domain $X$. Algorithm $\mathcal{A}$ is an ($\alpha, \beta, n$)-*PAC learner* for $C$ if for all $c \in C$ and all distributions $\mathcal{D}$ on $X$,

$$\Pr[(x_1, \ldots, x_n) \sim \mathcal{D}^n \,;\, h \sim \mathcal{A}((x_1, c(x_1)), \ldots, (x_n, c(x_n))) \,;\, \mathrm{error}_{\mathcal{D}}(c, h) \le \alpha] \ge 1 - \beta,$$

where the probability is over the sampling of $(x_1, \ldots, x_n)$ from $\mathcal{D}$ and the coin tosses of $\mathcal{A}$. The parameter $n$ is the *sample complexity* of $\mathcal{A}$.

See Appendix A for additional preliminaries on PAC learning.

## 2.3 Preliminaties from private learning

**Definition 2.12** (private PAC learning [Kasiviswanathan et al., 2011]). Algorithm $\mathcal{A}$ is a $(\alpha, \beta, \epsilon, \delta, n)$-private PAC learner if (i) $\mathcal{A}$ is an $(\alpha, \beta, n)$-PAC learner and (ii) $\mathcal{A}$ is $(\epsilon, \delta)$ differentially private.

Kasiviswanathan et al. [2011] provided a generic private learner with $O(\log(|X|))$ labeled samples.[7] Beimel et al. [2013a] introduced the representation dimension and showed that any concept class $C$ can be privately learned with $\Theta(\text{RepDim}(C))$ samples. For the sample complexity of $(\epsilon, \delta)$-differentially private learning of threshold functions over domain $X$, Bun et al. [2015] gave a lower bound of $\Omega(\log^* |X|)$. Recently, Cohen et al. [2022] gave a (nearly) matching upper bound of $\tilde{O}(\log^* |X|)$.

# 3 Towards private everlasting prediction

In this work, we extend private prediction to answering any sequence of prediction queries – unlimited in length. Our main goal is to present a generic private everlasting predictor with low training sample complexity $|S|$.

**Definition 3.1** (everlasting prediction). Let $\mathcal{A}$ be an algorithm with the following properties:

1. Algorithm $\mathcal{A}$ receives as input $n$ labeled examples $S = \{(x_i, y_i)\}_{i=1}^n \in (X \times \{0, 1\})^n$ and selects a hypothesis $h_0 : X \to \{0, 1\}$.

2. For round $r \in \mathbb{N}$, algorithm $\mathcal{A}$ gets a query, which is an unlabeled element $x_{n+r} \in X$, outputs $h_{r-1}(x_{n+r})$ and selects a hypothesis $h_r : X \to \{0, 1\}$.

We say that $\mathcal{A}$ is an $(\alpha, \beta, n)$-*everlasting predictor* for a concept class $C$ over a domain $X$ if the following holds for every concept $c \in C$ and for every distribution $\mathcal{D}$ over $X$. If $x_1, x_2, \ldots$ are sampled i.i.d. from $\mathcal{D}$, and the labels of the $n$ initial samples $S$ are correct, i.e., $y_i = c(x_i)$ for $i \in [n]$, then $\Pr[\exists r \geq 0 \text{ s.t. } \text{error}_\mathcal{D}(c, h_r) > \alpha] \leq \beta$, where the probability is over the sampling of $x_1, x_2, \ldots$ from $\mathcal{D}$ and the randomness of $\mathcal{A}$.

Applying the Dwork-Feldman notion of a private prediction interface to everlasting predictors we get:

**Definition 3.2.** An algorithm $\mathcal{A}$ is an $(\alpha, \beta, \epsilon, \delta, n)$-everlasting differentially private prediction interface if (i) $\mathcal{A}$ is a $(\epsilon, \delta)$-differentially private prediction interface $M$ (as in Definition 1.1), and (ii) $\mathcal{A}$ is an $(\alpha, \beta, n)$-everlasting predictor.

As a warmup, consider an $(\alpha, \beta, \epsilon, \delta, n)$-everlasting differentially private prediction interface $\mathcal{A}$ for concept class $C$ over (finite) domain $X$ (as in Definition 3.2 above). Assume that $\mathcal{A}$ does not vary its hypotheses, i.e. (in the language of Definition 3.1) $h_r = h_0$ for all $r > 0$.[8] Note that a computationally unlimited adversarial querying algorithm can recover the hypothesis $h_0$ by issuing all queries $x \in X$. Hence, in using $\mathcal{A}$ indefinitely we lose any potential benefits to sample complexity of restricting access to $h_0$ to being black-box and getting to the point where the lower-bounds on $n$ from private learning apply. A consequence of this simple observation is that a generic private everlasting predictor should answer all prediction queries with a single hypothesis – it should modify its hypothesis over time as it processes new queries.

We now take this observation a step further, showing that a private everlasting predictor that answers prediction queries solely based on its training sample $S$ (and randomness, but not on the queries) is subject to the same sample complexity lowerbounds as private learners.

---

[7]We omit the dependency on $\epsilon, \delta, \alpha, \beta$ in this brief review.

[8]Formally, $\mathcal{A}$ can be thought of as two mechanisms $\mathcal{A} = (M_0, M_1)$ where $M_0$ is $(\epsilon, \delta)$-differentially private. (i) On input a labeled training sample $S$ mechanism $M_0$ computes a hypothesis $h_0$. (ii) On a query $x \in X$ mechanism $M_1$ replies $h_0(x)$.

Consider an $(\alpha, \beta < 1/8, \epsilon, \delta, n)$-everlasting differentially private prediction interface $\mathcal{A}$ for concept class $C$ over (finite) domain $X$ that upon receiving the training set $S \in (X \times \{0,1\})^n$ selects an infinite sequence of hypotheses $\{h_r\}_{r \geq 0}$ where $h_r : X \to \{0,1\}$. Formally, we can think of $\mathcal{A}$ as composed of three mechanisms $\mathcal{A} = (M_0, M_1, M_2)$ where $M_0$ is $(\epsilon, \delta)$-differentially private:

- On input a labeled training sample $S \in (X \times \{0,1\})^n$ mechanism $M_0$ computes an initial state and an initial hypothesis $(\sigma_0, h_0) = M_0(S)$.

- On a query $x_{n+r}$ mechanism $M_1$ produces an answer $M_1(x_{n+r}) = h_i(x_{n+r})$ and mechanism $M_2$ updates the hypothesis-state pair $(h_{r+1}, \sigma_{r+1}) = M_2(\sigma_r)$.

Note that as $M_0$ and $M_2$ do not receive the sequence $\{x_{n+r}\}_{r \geq 0}$ as input, the sequence $\{h_r\}_{r \geq 0}$ depends solely on $S$. Furthermore as $M_1$ and $M_2$ post-process the outcome of $M_0$, i.e., the sequence of queries and predictions $\{(x_r, h_r(x_r))\}_{r \geq 0}$ preserves $(\epsilon, \delta)$-differential privacy with respect to the training set $S$. In Appendix B we prove:

**Theorem 3.3.** $\mathcal{A}$ can be transformed into a $(O(\alpha), O(\beta), \epsilon, \delta, O(n \log(1/\beta)))$-private PAC learner for $C$.

# 4 Private everlasting prediction – definition

Theorem 3.3 requires us to seek private predictors whose prediction relies on more information than what is provided by the initial labeled sample. Possibilities include requiring the input of additional labeled or unlabeled examples during the lifetime of the predictor, while protecting the privacy of these examples.

We choose to rely on the queries for updating the predictor's internal state. This introduces a potential privacy risk for these queries as sensitive information about a query may be leaked in the predictions following it. Furthermore, we need take into account that a privacy attacker may choose their queries adversarially and adaptively.

**Definition 4.1** (private everlasting black-box prediction). An algorithm $\mathcal{A}$ is an $(\alpha, \beta, \varepsilon, \delta, n)$-private everlasting black-box predictor for a concept class $C$ if

1. **Prediction:** $\mathcal{A}$ is an $(\alpha, \beta, n)$-everlasting predictor for $C$ (as in Definition 3.1).

2. **Privacy:** For every adversary $\mathcal{B}$ and every $t \geq 1$, the random variables $\text{View}^0_{\mathcal{B},t}$ and $\text{View}^1_{\mathcal{B},t}$ (defined in Figure 1) are $(\varepsilon, \delta)$-indistinguishable.

# 5 Tools from prior works

We briefly describe tools from prior works that we use in our construction. See Appendix C for a more detailed account.

**Algorithm LabelBoost [Beimel et al., 2021]:** Algorithm LabelBoost takes as input a partially labeled database $S \circ T \in (X \times \{0, 1, \perp\})^*$ (where the first portion of the database, $S$, contains examples by some concept $c \in C$) and outputs a similar database where both $S$ and $T$ are (re)labeled by a concept $h \in C$ such that $\text{error}_{\mathcal{D}}(c, h)$ is bounded. We use the following lemmata from Beimel et al. [2021]:

**Lemma 5.1** (privacy of Algorithm LabelBoost). *Let $\mathcal{A}$ be an $(\epsilon, \delta)$-differentially private algorithm operating on labeled databases. Construct an algorithm $\mathcal{B}$ that on input a partially labeled database $S \circ T \in (X \times \{0, 1, \perp\})^*$ applies $\mathcal{A}$ on the outcome of LabelBoost$(S \circ T)$. Then, $\mathcal{B}$ is $(\epsilon + 3, 4e\delta)$-differentially private.*

**Lemma 5.2** (Utility of Algorithm LabelBoost). *Fix $\alpha$ and $\beta$, and let $S \circ T$ be s.t. $S$ is labeled by some target concept $c \in C$, and s.t. $|T| \leq \frac{\beta}{e} \text{VC}(C) \exp(\frac{\alpha |S|}{2\text{VC}(C)}) - |S|$. Consider the execution of LabelBoost on $S \circ T$, and let $h$ denote the hypothesis chosen by LabelBoost to relabel $S \circ T$. With probability at least $(1 - \beta)$ we have that $\text{error}_S(h) \leq \alpha$.*

**Parameters:** $b \in \{0,1\}$, $t \in \mathbb{N}$.

**Training Phase:**

1. The adversary $\mathcal{B}$ chooses two sets of $n$ labeled elements $(x_1^0, y_1^0), \ldots, (x_n^0, y_n^0)$ and $(x_1^1, y_1^1), \ldots, (x_n^1, y_n^1)$, subject to the restriction $\left| \{ i \in [n] : (x_i^0, y_i^0) \neq (x_i^1, y_i^1) \} \right| \in \{0, 1\}$.

2. If $\exists i$ s.t. $(x_i^0, y_i^0) \neq (x_i^1, y_i^1)$ then set Flag = 1. Otherwise set Flag = 0.

3. Algorithm $\mathcal{A}$ gets $(x_1^b, y_1^b), \ldots, (x_n^b, y_n^b)$ and selects a hypothesis $h_0 : X \to \{0, 1\}$.
   \* the adversary $\mathcal{B}$ does not get to see the hypothesis $h_0$ *\

**Prediction phase:**

4. For round $r = 1, 2, \ldots, t$:

   (a) If Flag = 1 then the adversary $\mathcal{B}$ chooses two elements $x_{n+r}^0 = x_{n+r}^1 \in X$. Otherwise, the adversary $\mathcal{B}$ chooses two elements $x_{n+r}^0, x_{n+r}^1 \in X$.

   (b) If $x_{n+r}^0 \neq x_{n+r}^1$ then Flag is set to 1.

   (c) If $x_{n+r}^0 = x_{n+r}^1$ then the adversary $\mathcal{B}$ gets $h_{r-1}(x_{n+r}^b)$.
   \* the adversary $\mathcal{B}$ does not get to see the label if $x_{n+r}^0 \neq x_{n+r}^1$ *\

   (d) Algorithm $\mathcal{A}$ gets $x_{n+r}^b$ and selects a hypothesis $h_r : X \to \{0, 1\}$.
   \* the adversary $\mathcal{B}$ does not get to see the hypothesis $h_r$ *\

   Let $\text{View}_{\mathcal{B},t}^b$ be $\mathcal{B}$'s view of the execution, i.e., its own randomness and the sequence of predictions in Step 4c.

Figure 1: Definition of $\text{View}_{\mathcal{B},t}^0$ and $\text{View}_{\mathcal{B},t}^1$.

# 6    A Generic Construction

Our generic construction Algorithm `GenericBBL` transforms a (non-private) learner for a concept class $C$ into a private everlasting predictor for $C$. The theorem below follows from Theorem 6.2 and Claim 6.3 which are proved in Appendix E.

**Theorem 6.1.** *Given $\alpha, \beta, \delta < 1/16, \epsilon < 1$, Algorithm `GenericBBL` is a $(4\alpha, 4\beta, \epsilon, \delta, n)$-private everlasting predictor, where $n$ is set as in Algorithm `GenericBBL`.*

**Theorem 6.2** (accuracy of algorithm `GenericBBL`)**.** *Given $\alpha, \beta, \delta < 1/16$, $\varepsilon < 1$, for any concept $c$ and any round $r$, algorithm `GenericBBL` can predict the label of $x_r$ as $h_r(x_r)$, such that $\Pr[error_{\mathcal{D}}(c(x_r) \neq h_r(x_r)) \leq 4\alpha] \geq 1 - 4\beta$.*

**Claim 6.3** (privacy of algorithm `GenericBBL`)**.** *`GenericBBL` is $(\epsilon, \delta)$-differentially private.*

**Remark 6.4.** *For simplicity, we analyzed `GenericBBL` in the realizable setting, i.e., under the assumption that the training set $S$ is* consistent *with the target class $C$. Our construction carries over to the agnostic setting via standard arguments (ignoring computational efficiency). We refer the reader to [Beimel et al., 2021] and [Alon et al., 2020] for generic agnostic-to-realizable reductions in the context of private learning.*

## 6.1    Improving the sample complexity dependency on accuracy

We briefly sketch how to improve the sample complexity of Algorithm `GenericBBL` from $n = \tilde{\Theta}\left(\frac{\text{VC}^2(C)}{\alpha^2 \epsilon^2}\right)$ to $n = \tilde{\Theta}\left(\frac{\text{VC}^2(C)}{\alpha \epsilon^2}\right)$ by modifying steps 3(d)ii and 3(d)iii of Algorithm `GenericBBL`. To simplify the description, we consider a constant $\epsilon$, we ignore the privacy amplification by subsampling occurring in steps 2 and 3f and illustrate how the modified algorithm would execute by considering round $i = 1$ of Algorithm `GenericBBL`.

Note that $S_1 = S$ where $n = |S_1| = T_1 \cdot \lambda_1$ and $\lambda_1 = \tilde{\Theta}(\text{VC}(C)/\alpha_1)$ is the sample complexity needed such that each of the hypotheses computed in Step 3b has error $\alpha_1$ except for

---

**Algorithm** `GenericBBL`

---

**Initial input:** A labeled database $S \in (X \times \{0,1\})^n$ where $n = \frac{8\tau}{\alpha^2 \varepsilon^2} \cdot \left(8\mathrm{VC}(C)\log(\frac{26}{\alpha}) + 4\log(\frac{4}{\beta})\right)^2 \cdot \log(\frac{1}{\delta}) \cdot \log^2\left(\frac{64\mathrm{VC}(C)\log(\frac{26}{\alpha}) + 32\log(\frac{4}{\beta})}{\varepsilon \alpha^2 \beta \delta}\right) \cdot (3 + \exp(\varepsilon + 4))$.

1. Let $\tau > 50$. Set $\alpha_1 = \alpha/2$, $\beta_1 = \beta/2$. Define $\lambda_i = \frac{8\mathrm{VC}(C)\log(\frac{13}{\alpha_i}) + 4\log(\frac{2}{\beta_i})}{\alpha_i}$.
   /* by Theorem A.2 $\lambda_i$ samples suffice for PAC learning $C$ with parameters $\alpha_i, \beta_i$ */

2. Let $S_1 \subseteq S$ be a random subset of size $n \cdot \frac{\varepsilon}{3+\exp(\varepsilon+4)} = \frac{\tau \cdot \lambda_i^2 \cdot \log(\frac{1}{\delta}) \cdot \log^2(\frac{\lambda_i}{\varepsilon \alpha_i \beta_i \delta})}{\varepsilon}$.

3. Repeat for $i = 1, 2, 3, \ldots$

   (a) Divide $S_i$ into $T_i = \frac{\tau \cdot \lambda_i \cdot \log(\frac{1}{\delta}) \cdot \log^2(\frac{\lambda_i}{\varepsilon \alpha_i \beta_i \delta})}{\varepsilon}$ disjoint databases $S_{i,1}, \ldots, S_{i,T_i}$ of size $\lambda_i$.

   (b) For $t \in [T_i]$ let $f_t \in C$ be a hypothesis minimizing error$_{S_{i,t}}(\cdot)$. Define $F_i = (f_1, \ldots, f_{T_i})$.

   (c) Set $R_i = \frac{12800|S_i|}{\varepsilon}$. Set the privacy parameters $\varepsilon_i' = \frac{1}{3\sqrt{R_i \ln(\frac{2}{\delta})}}$ and $\delta_i' = \frac{\delta}{2R_i}$. Instantiate noises $\eta_1, \ldots, \eta_{R_i} \sim \mathrm{Lap}(1/T_i \varepsilon_i')$.

   (d) For $\ell = 1$ to $R_i$:
   
       i. Receive as input a prediction query $x_{i,\ell} \in X$.
       ii. Compute $q_{x_{i,\ell}}(F_i) = \frac{1}{T_i} \sum_{t \in [T_i]} f_t(x_{i,\ell})$, and let $y_{i,\ell} = q_{x_{i,\ell}}(F_i) + \eta_\ell$.
       iii. Respond with the label 0 if $y_{i,\ell} < 1/2$ and 1 if $y_{i,\ell} \geq 1/2$.

   (e) Denote $D_i = (x_{i,1}, \ldots, x_{i,R_i})$.

   (f) Let $\hat{S}_i \subseteq S_i$ and $\hat{D}_i \subseteq D_i$ be random subsets of size $\frac{\varepsilon|S_i|}{3+\exp(\varepsilon+4)}$ and $\frac{\varepsilon|D_i|}{3+\exp(\varepsilon+4)}$ respectively, and let $\hat{S}_i' \circ \hat{D}_i' \leftarrow \mathtt{LabelBoost}(\hat{S}_i \circ \hat{D}_i)$. Let $S_{i+1} \subseteq \hat{D}_i'$ be a random subset of size $\lambda_{i+1} T_{i+1}$.

   (g) Set $\alpha_{i+1} \leftarrow \alpha_i/2$ and $\beta_{i+1} \leftarrow \beta_i/2$.

---

probability $\beta_i$. Applying advanced composition, we get that $\tilde{\Theta}(T_1^2)$ noisy majority queries implemented in steps steps 3(d)ii and 3(d)iii can be performed, i.e., $R_1 = \tilde{\Theta}(T_1^2)$. Finally, to feed the next phase with $|S_2| = \Theta(|S_1|)$ labeled samples, we need that $R_1 = \Theta(T_1 \cdot \lambda_1) = \tilde{\Theta}(T_1 \cdot \mathrm{VC}(C)/\alpha)$ and hence $T_1 = \tilde{\Theta}(VC(C)/\alpha)$ resulting in $n = \tilde{\Theta}(\mathrm{VC}^2(C)/\alpha^2)$.

However, when queries are made from the same underlying distribution $S$ was selected, we expect that most of them would exhibit a clear majority in steps 3(d)ii and 3(d)iii, except for a fraction of $O(\alpha)$. Hence a natural way to improve on the number of majority queries that can be performed is to replace steps 3(d)ii and 3(d)iii with a decision based on the sparse vector technique of Dwork et al. [2009]. In particular, we can use the `BetweenThresholds` mechanism of Bun et al. [2016] to get an improved $R_1 = \tilde{\Theta}(T_1^2/\alpha)$ and hence get $T_1 = \tilde{\Theta}(VC(C))$ and $n = \tilde{\Theta}(VC^2/\alpha)$.

A final important wrinkle is that the above calculation is based on the queries coming from the same underlying distribution $S$ was selected, while our worst-case privacy requirement allows for an adversarial choice of queries that may in turn cause the execution of `BetweenThresholds` to halt too often and hence exhaust the privacy budget for the phase. It is hence important to estimate the number of times `BetweenThresholds` halts within a phase and to stop the execution of Algorithm `GenericBBL` when the estimate crosses a threshold. This estimate needs to be done in an online manner and should preserve differential privacy with respect to the queries.[9] This can be done, e.g., using the *private counter* algorithm of Dwork et al. [2010a], which preserves differential privacy under continual observation.

---

[9]For example, stopping `GenericBBL` after the count of halts crosses a precise threshold would reveal that the last query caused `BetweenThresholds` to halt, and hence breach differential privacy.

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

# A Additional Preliminaries from PAC Learning

It is well know that that a sample of size $\Theta(\text{VC}(C))$ is necessary and sufficient for the PAC learning of a concept class $C$, where the Vapnik-Chervonenkis (VC) dimension of a class $C$ is defined as follows:

**Definition A.1** (VC-Dimension [Vapnik and Chervonenkis, 1971]). Let $C$ be a concept class over a domain $X$. For a set $B = \{b_1, \ldots, b_\ell\} \subseteq X$ of $\ell$ points, let $\Pi_C(B) = \{(c(b_1), \ldots, c(b_\ell)) : c \in C\}$ be the set of all dichotomies that are realized by $C$ on $B$. We say that the set $B \subseteq X$ is *shattered* by $C$ if $C$ realizes all possible dichotomies over $B$, in which case we have $|\Pi_C(B)| = 2^{|B|}$.

The VC dimension of the class $C$, denoted $\text{VC}(C)$, is the cardinality of the largest set $B \subseteq X$ shattered by $C$.

**Theorem A.2** (VC bound). *Let $C$ be a concept class over a domain $X$. For $\alpha, \beta < 1/2$, there exists an $(\alpha, \beta, n)$-PAC learner for $C$, where $n = \frac{8\text{VC}(C)\log(\frac{13}{\alpha}) + 4\log(\frac{2}{\beta})}{\alpha}$.*

# B Proof of Theorem 3.3

The proof of Theorem 3.3 follows from algorithms `HypothesisLearner`, `AccuracyBoost` and claims B.1, B.2, all described below.

In Algorithm `HypothesisLearner` we assume that the everlasting differentially private prediction interface $\mathcal{A}$ was fed with $n$ i.i.d. samples taken from some (unknown) distribution $\mathcal{D}$ and labeled by an unknown concept $c \in C$. Assuming the sequence of hypotheses $\{h_r\}_{r \geq 0}$ produced by $\mathcal{A}$ satisfies

$$\forall r \quad \text{error}_{\mathcal{D}}(c, h_r) \leq \alpha \tag{1}$$

we use it to construct – with constant probability – a hypothesis $h$ with error bounded by $O(\alpha)$.

---

**Algorithm** `HypothesisLearner`

---

**Parameters:** $0 < \beta \leq 1/8$, $R = |X| \cdot \log(|X|) \cdot \log(1/\beta)$
**Input:** hypothesis sequence $\{h_r\}_{r \geq 0}$

1. for all $x \in X$ let $L_x = \emptyset$

2. for $r = 0, 1, 2, \ldots, R$
   (a) select $x$ uniformly at random from $X$ and let $L_x = L_x \cup \{h_r(x)\}$

3. if $L_x = \emptyset$ for some $x \in X$ then fail, output an arbitrary hypothesis, and halt
   /* $\Pr[\exists x \text{ such that } L_x = \emptyset] \leq |X|(1 - \frac{1}{|X|})^R \approx |X|e^{-R/|X|} = \beta$ */

4. for all $x \in X$ let $r_x$ be sampled uniformly at random from $L_x$

5. construct the hypothesis $h$, where $h(x) = r_x$

---

**Claim B.1.** *If executed on a hypothesis sequence satisfying Equation 1 then with probability at least 3/4 Algorithm `HypothesisLearner` outputs a hypothesis $h$ satisfying $\text{error}_{\mathcal{D}}(c, h) \leq 8\alpha$.*

*Proof.* Having $\mathcal{D}, c \in C$ fixed, and given a hypothesis $h$, we define $e_h(x)$ to be 1 if $h(x) \neq c(x)$ and 0 otherwise. Thus, we can write $\text{error}_{\mathcal{D}}(c, h) = \mathbb{E}_{x \sim \mathcal{D}}[e_h(x)]$.

Observe that when Algorithm `HypothesisLearner` does not fail, $r_x$ (and hence $h(x)$) is chosen with equal probability among $(h_1(x), h_2(x), \ldots, h_R(x))$ and hence $\mathbb{E}_\theta[e_h(x)] = \mathbb{E}_{i \in_R [R]}[e_{h_i}(x)]$ where $\theta$ denotes the randomness of `HypothesisLearner`. We get:

$$
\begin{aligned}
\mathbb{E}_\theta[\text{error}_{\mathcal{D}}(c, h)] &= \mathbb{E}_\theta \mathbb{E}_{x \sim \mathcal{D}}[e_h(x)] = \mathbb{E}_{x \sim \mathcal{D}} \mathbb{E}_\theta[e_h(x)] \\
&= \mathbb{E}_{x \sim \mathcal{D}} \mathbb{E}_{i \in_R [R]}[e_{h_i}(x)] = \mathbb{E}_{i \in_R [R]} \mathbb{E}_{x \sim \mathcal{D}}[e_{h_i}(x)] \\
&\leq \mathbb{E}_{i \sim \mathcal{R}}[\alpha] = \alpha.
\end{aligned}
$$

By Markov inequality, we have $\Pr_\theta[\text{error}_{\mathcal{D}}(c, h) \geq 8\alpha] \leq 1/8$. The claim follows noting that Algorithm `HypothesisLearner` fails with probability at most $\beta \leq 1/8$. □

The second part of the transformation is Algorithm `AccuracyBoost` that applies Algorithm `HypothesisLearner` $O(\log(1/\beta))$ times to obtain with high probability a hypothesis with $O(\alpha)$ error.

---

**Algorithm `AccuracyBoost`**

---

**Parameters:** $\beta$, $R = 104 \ln \frac{1}{\beta}$
**Input:** $R$ labeled samples with $n$ examples each $(S_1, \ldots, S_R)$ where $S_i \in (X \times \{0,1\})^n$

1. for $i = 1, 2 \ldots R$
   (a) execute $\mathcal{A}(S_i)$ to obtain a hypothesis sequence $\{h_r^i\}_{r \geq 0}$
   (b) execute Algorithm `WeakHypothesisLearner` on $\{h_r^i\}_{r \geq 0}$ to obtain hypothesis $h^i$
2. construct the hypothesis $\hat{h}$, where $\hat{h}(x) = \text{maj}(h^1(x), \ldots, h^R(x))$.

---

**Claim B.2.** *With probability $1 - \beta$, Algorithm `AccuracyBoost` output a $24\alpha$-good hypothesis over distribution $\mathcal{D}$.*

*Proof.* Define $B_i$ to be the event where the sequence of hypotheses $\{h_r^i\}_{r \geq 0}$ produced in Step 1a of `AccuracyBoost` does not satisfy Equation 1. We have,

$$\Pr[\text{error}_{\mathcal{D}}(c, h_i) > 8\alpha] \leq \Pr[B] + (1 - \Pr[B]) \cdot \Pr[\text{error}_{\mathcal{D}}(c, h) > 8\alpha] \leq \beta + 1/4 < 3/8.$$

Hence, by the Chernoff bound, when $R \geq 104 \ln \frac{1}{\beta}$, we have at least $7R/8$ hypotheses are $8\alpha$-good over distribution $\mathcal{D}$. Consider the worst case, in which $R/8$ hypotheses always output wrong labels. To output a wrong label of $x$, we require at least $3R/8$ hypotheses to output wrong labels. Thus $h$ is $24\alpha$-good over distribution $\mathcal{D}$. $\qquad\square$

## C  Tools from Prior Works

### C.1  Algorithm `LabelBoost` [Beimel et al., 2021]

---

**Algorithm `LabelBoost` [Beimel et al., 2021]**

---

**Parameters:** A concept class $C$.
**Input:** A partially labeled database $S \circ T \in (X \times \{0,1,\bot\})^*$.
% We assume that the first portion of the database (denoted $S$) contains labeled examples. The algorithm outputs a similar database where both $S$ and $T$ are (re)labeled.
1. Initialize $H = \emptyset$.
2. Let $P = \{p_1, \ldots, p_\ell\}$ be the set of all points $p \in X$ appearing at least once in $S \circ T$. Let $\Pi_C(P) = \{(c(p_1), \ldots, c(p_\ell)) : c \in C\}$ be the set of all dichotomies generated by $C$ on $P$.
3. For every $(z_1, \ldots, z_\ell) \in \Pi_C(P)$, add to $H$ an arbitrary concept $c \in C$ s.t. $c(p_i) = z_i$ for every $1 \leq i \leq \ell$.
4. Choose $h \in H$ using the exponential mechanism with privacy parameter $\epsilon = 1$, solution set $H$, and the database $S$.
5. (Re)label $S \circ T$ using $h$, and denote the resulting database $(S \circ T)^h$, that is, if $S \circ T = (x_i, y_i)_{i=1}^t$ then $(S \circ T)^h = (x_i, y_i')_{i=1}^t$ where $y_i' = h(x_i)$.
6. Output $(S \circ T)^h$.

---

**Lemma C.1** (privacy of Algorithm `LabelBoost` [Beimel et al., 2021])**.** *Let $\mathcal{A}$ be an $(\epsilon, \delta)$-differentially private algorithm operating on partially labeled databases. Construct an algorithm $\mathcal{B}$ that on input a partially labeled database $S \circ T \in (X \times \{0,1,\bot\})^*$ applies $\mathcal{A}$ on the outcome of `LabelBoos`($S \circ T$). Then, $\mathcal{B}$ is $(\epsilon + 3, 4e\delta)$-differentially private.*

Consider an execution of `LabelBoost` on a database $S \circ T$, and assume that the examples in $S$ are labeled by some target concept $c \in C$. Recall that for every possible labeling $\vec{z}$ of the elements in $S$ and in $T$, algorithm `LabelBoost` adds to $H$ a hypothesis from $C$ that agrees with $\vec{z}$. In particular, $H$ contains a hypothesis that agrees with the target concept $c$ on $S$ (and on $T$). That is, $\exists f \in H$ s.t. $\mathrm{error}_S(f) = 0$. Hence, the exponential mechanism (on Step 4) chooses (w.h.p.) a hypothesis $h \in H$ s.t. $\mathrm{error}_S(h)$ is small, provided that $|S|$ is roughly $\log|H|$, which is roughly $\mathrm{VC}(C) \cdot \log(|S| + |T|)$ by Sauer's lemma. So, algorithm `LabelBoost` takes an input database where only a small portion of it is labeled, and returns a similar database in which the labeled portion grows exponentially.

**Lemma C.2** (utility of Algorithm `LabelBoost` [Beimel et al., 2021]). *Fix $\alpha$ and $\beta$, and let $S \circ T$ be s.t. $S$ is labeled by some target concept $c \in C$, and s.t.*

$$|T| \le \frac{\beta}{e} \mathrm{VC}(C) \exp\left(\frac{\alpha|S|}{2\mathrm{VC}(C)}\right) - |S|.$$

*Consider the execution of `LabelBoost` on $S \circ T$, and let $h$ denote the hypothesis chosen on Step 4. With probability at least $(1 - \beta)$ we have that $\mathrm{error}_S(h) \le \alpha$.*

# D   Some Technical Facts

We refer to the execution of steps 3a-3g of algorithm `GenericBBL` as a *phase* of the algorithm, indexed by $i = 1, 2, 3, \dots$.

In `GenericBBL`, we require that the set of all predictions in the phase $i$ is $(1, \delta)$-differentially private.

**Claim D.1.** *For $\delta < 1$, the composited Laplace mechanism used in step 3c in the $i$-th iteration, is $(1, \delta)$-differentially private.*

*Proof.* Let $\varepsilon_i', \delta_i'$ be as in Step 3c. Since $e^{\varepsilon_i'} - 1 < 2\varepsilon_i'$ for $0 < \varepsilon_i' < 1$, we have

$$\sqrt{2R_i \ln\left(\frac{1}{R_i \delta_i'}\right)} \cdot \varepsilon_i' + R_i \varepsilon_i'(e^{\varepsilon_i'} - 1) \le \sqrt{2R_i \ln\left(\frac{2}{\delta}\right)} \cdot \varepsilon_i' + 2R_i \varepsilon_i'^2 = \frac{\sqrt{2}}{3} + \frac{2}{9\ln(\frac{2}{\delta})} \le 1.$$

The proof is concluded by using advanced composition (Theorem 2.6). $\qquad\square$

In step 3f, `GenericBBL` takes a random subset of size $\lambda_{i+1} T_{t+1}$ from $\hat{D}_i'$. We show that the size of $\hat{D}_i'$ is at least $\lambda_{i+1} T_{t+1}$.

**Claim D.2.** *When $\varepsilon \le 1$, for any $i \ge 1$, we always have $|\hat{D}_i'| \ge \lambda_{i+1} T_{i+1}$.*

*Proof.* Let $m = 3 + \exp(\varepsilon + 4) < 200$. By the step 3c, step 3e and step 3f, $|\hat{D}_j| = \frac{\varepsilon|D_j|}{m} = \frac{12800|S_j|}{m} \ge 64|S_j| = 64\lambda_j T_j$. Then it is sufficient to verify $128\lambda_j T_j \ge \lambda_{j+1} T_{j+1}$

We can verify that

$$4\lambda_j = 4 \cdot \frac{8\mathrm{VC}(C)\log(\frac{13}{\alpha_i}) + 4\log(\frac{2}{\beta_i})}{\alpha_i} = 4 \cdot \frac{8\mathrm{VC}(C)(\log(\frac{13}{\alpha_{j+1}}) - 1) + 4(\log(\frac{2}{\beta_{j+1}}) - 1)}{2\alpha_{j+1}} \ge \lambda_{j+1}$$

and

$$16T_j = \frac{16\tau \cdot \lambda_i \cdot \log(\frac{1}{\delta}) \cdot \log^2(\frac{\lambda_i}{\varepsilon\alpha_i\beta_i\delta})}{\varepsilon} \ge \frac{4\tau \cdot \lambda_{i+1} \cdot \log(\frac{1}{\delta}) \cdot \log^2(\frac{\lambda_{i+1}}{16\varepsilon\alpha_{i+1}\beta_{i+1}\delta})}{\varepsilon} \ge T_{j+1}.$$

The last inequalitu holds because $\lambda_j \ge 4$ and $\alpha_j, \beta_j \le 1/2$. $\qquad\square$

To apply the privacy and accuracy of *LabelBoost*, the sizes of the databases need to satisfy the inequalities in lemma C.2. We verify that in each phase, the sizes of the databases always satisfy the requirement.

**Claim D.3.** *When $\varepsilon \leq 1$, for any $i \geq 1$, we have $|\hat{D}_i| \leq \frac{\beta_i}{e} VC(C) exp\left(\frac{\alpha_i |\hat{S}_i|}{2VC(C)}\right) - |\hat{S}_i|$.*

*Proof.* By claim D.2, step 3c and step 3f,

$$|\hat{D}_i| = \frac{\varepsilon |D_i|}{m} = O(\lambda_i T_i) = O\left(VC(C)\log^2(VC(C)) \cdot poly\left(\frac{1}{\alpha_i}, \log(\frac{1}{\beta_i}), \frac{1}{\varepsilon}, \log(\frac{1}{\delta})\right)\right)$$

and

$$
\begin{aligned}
|\hat{S}_i| &= \frac{\varepsilon |S_i|}{m} \\
&= O(\varepsilon \lambda_i T_i) = O(\lambda_i T_i) \\
&= O\left(VC(C)\log^2(VC(C)) \cdot poly\left(\frac{1}{\alpha_i}, \log(\frac{1}{\beta_i}), \frac{1}{\varepsilon}, \log(\frac{1}{\delta})\right)\right).
\end{aligned}
\tag{2}
$$

Note that

$$\frac{\beta_i}{e} VC(C) exp\left(\frac{\alpha_i |\hat{S}_i|}{2VC(C)}\right) = \Omega\left(VC^2(C) \cdot exp\left(poly\left(\frac{1}{\alpha_i}, \log(\frac{1}{\beta_i}), \frac{1}{\varepsilon}, \log(\frac{1}{\delta})\right)\right)\right),$$

for $T_i = \frac{\tau \cdot \lambda_i \cdot \log(\frac{1}{\delta}) \cdot \log^2(\frac{\lambda_i}{\varepsilon \alpha_i \beta_i \delta})}{\varepsilon}$, the inequality holds when $\tau \geq 50$. $\square$

## E   Accuracy of Algorithm `GenericBBL` – proof of Theorem 6.2

We refer to the execution of steps 3a-3g of algorithm `GenericBBL` as a *phase* of the algorithm, indexed by $i = 1, 2, 3, \ldots$.

We give some technical facts in Appendix D. In Claim E.1, we show that in each phase, samples are labeled with high accuracy. In Claim E.3, we prove that algorithm `GenericBBL` fails with low probability. In Claim E.4, we prove that algorithm `GenericBBL` predict the labels with high accuracy.

**Claim E.1.** *For each phase $i$ we have*

$$\Pr\left[\exists g_i \in C \text{ s.t. } error_{S_i}(g) = 0 \text{ and } error_{\mathcal{D}}(g_i, c) \leq \sum_{j=1}^{i} \alpha_j\right] \geq 1 - 2\sum_{j=0}^{i} \beta_j.$$

*Proof.* The proof is by induction on $i$. The base case for $i = 1$ is trivial, with $g_1 = c$. Assume the claim holds for all $j \leq i$. By the properties of `LabelBoost` (Lemma C.2) and Claim D.3, with probability at least $1 - \beta_i$ we have that $S_i$ is labeled by a hypothesis $g_i \in C$ s.t. $error_{S_{i-1}}(g_{i-1}, g_i) \leq \alpha_i$. Observe that the points in $S_i$ (without their labels) are chosen i.i.d. from $\mathcal{D}$, and hence, By Theorem A.2 (VC bounds) and $|S_{i-1}| \geq 128\lambda_{i-1} \geq \lambda_i$, with probability at least $1 - \beta_i$ we have that $error_{\mathcal{D}}(g_i, g_{i-1}) \leq \alpha_i$. Hence, with probability $1 - 2\beta_i$, we have $error_{\mathcal{D}}(g_i, g_{i-1}) \leq \alpha_i$. Finally, by the triangle inequality, $error_{\mathcal{D}}(g_i, c) \leq \sum_{j=1}^{i} \alpha_j$, except with probability $2\sum_{j=1}^{i} \beta_j$ $\square$

Combining claims E.1 union bound, we get:

**Claim E.2.** *Let $\mathcal{D}$ be an underlying distribution and let $c \in C$ be a target concept. Then*

$$\Pr[\forall i \ \exists g_i \in C \text{ s.t. } error_{S_i}(g_i) = 0 \text{ and } error_{\mathcal{D}}(g_i, c) \leq \alpha] \geq 1 - 2\beta.$$

For each phase $i$, if there exists a noise $\eta_j \geq 1/16$ in step 3c, we say the phase $i$ fails. Define the following good event.

> **Event $E_1$:** Phase $i$ doesn't fail for all $i \geq 1$.

**Claim E.3.** *When $\lambda \geq 1, \varepsilon \leq 1, \alpha \leq 1/2, \beta \leq 1/2, \delta \leq 1/2$, event $E_1$ occurs with probability at least $1 - \beta$.*

*Proof.* In each phase $i$, For each $\eta_j$, by the property of Laplace distribution, we have $\Pr[|\eta| \geq 1/16] = e^{-T_i \varepsilon'_i/16} = e^{-\frac{T_i}{48\sqrt{R_i \ln(2/\delta)}}} \leq \beta_i/R_i$. This inequality holds when $\tau \geq 1$. By union bound, we have $\Pr[\text{phase } i \text{ fails}] \leq \beta_i$.

Using to union bound,
$$\Pr[\text{Event } E_1 \text{ occurs}] \geq 1 - \beta.$$

$\square$

**Notations.** Consider the $i$th phase of Algorithm `GenericBBL`, and focus on the $j$-th iteration of Step 3. Fix all of the randomness of noises. Now observe that the output on step 3(d)iii is a deterministic function of the input $x_{i,j}$. This defines a hypothesis which we denote as $h_{i,j}$.

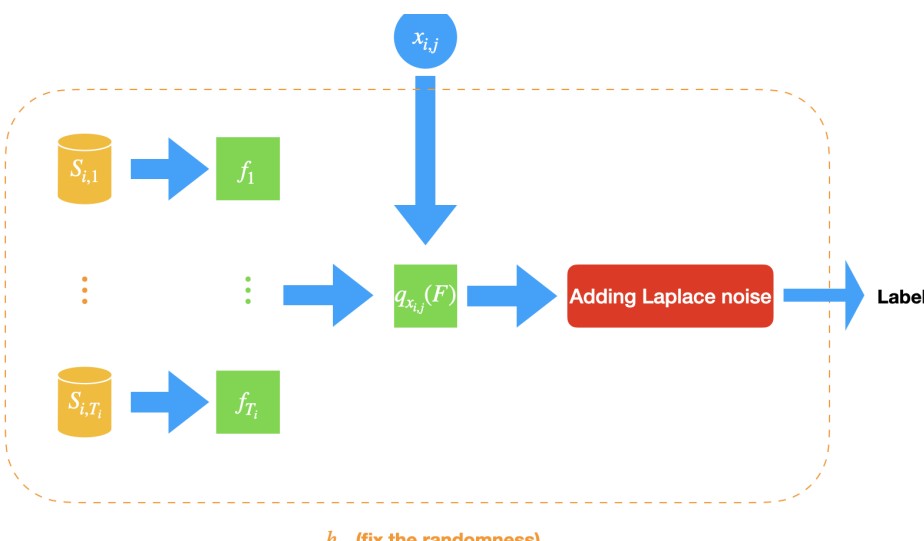

Figure 2: Hypothesis $h_{i,j}$

**Claim E.4.** *For $\beta < 1/16$, with probability at least $1 - 4\beta$, all of the hypotheses defined above are $4\alpha$-good w.r.t. $\mathcal{D}$ and $c$.*

*Proof.* In the phase $i$, by Claim E.2, with probability at least $1 - 2\beta$ we have that $S_i$ is labeled by a hypothesis $g_i \in C$ satisfying $\text{error}_{\mathcal{D}}(g_i, c) \leq \alpha$. We continue with the analysis assuming that this is the case.

On step 3a of the $i$th phase we divide $S_i$ into $T_i$ subsamples of size $\lambda_i$ each, identify a consistent hypothesis $f_t \in C$ for every subsample $S_{i,t}$, and denote $F_i = (f_1, \dots, f_T)$. By Theorem A.2 (VC bounds), every hypothesis in $F_i$ satisfies $\text{error}_{\mathcal{D}}(f_t, g_i) \leq \alpha$ with probability $1 - \beta_i$, in which case, by the triangle inequality we have that $\text{error}_{\mathcal{D}}(f_t, c) \leq 2\alpha$.

Set $T_i \geq \frac{512(1-4\beta_i)\ln(\frac{1}{\beta_i})}{(1-64\beta_i)^2}$, using Chernoff bound, it holds that for at least $15T_i/16$ of the hypotheses in $F_i$ have error $\text{error}_{\mathcal{D}}(f_t, c) \leq 2\alpha$ with probability at least $1 - \beta_i$.

By Claim E.3, with probability $1 - \beta$, all Laplace noises added in step 3d is less than $\frac{T_i}{16}$. Let $m : X \to \{0,1\}$ defined as $m(x) = \text{maj}_{f_t \in F_i}(f_t(x))$. For $m$ to err on a point $x$ (w.r.t. the target concept $c$), it must be that at least 3/8-fraction of the $2\alpha$-good hypotheses in $\hat{F}_i$ err on $x$. Consider the worst case in Figure 3, based on Claim E.3, we have $\text{error}_{\mathcal{D}}(m, c) \leq 4\alpha$ with probability $1 - \beta$.

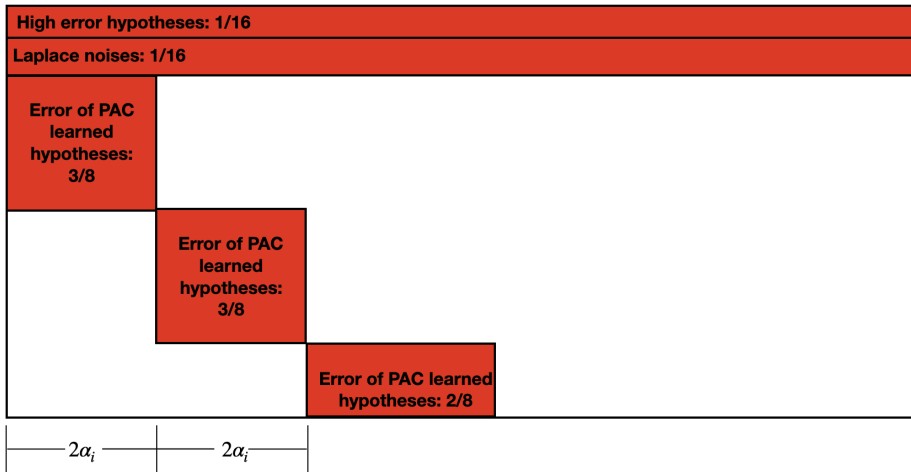

Figure 3: The horizontal represents the input point. The vertical represents the hypothesis. The red parts represent the incorrect prediction. We let $\frac{T_i}{16}$ hypothesis predict all labels incorrectly and let Laplace noises to be as large as $\frac{T_i}{16}$. To output an incorrect label, there must exist $\frac{3T_i}{8}$ hypothesis output the incorrect label. In the worst case, at most $4\alpha$ of points are incorrectly classified.

Conclude all above, with probability $1 - 4\beta$, all the hypotheses are $4\alpha$-good. $\qquad\square$

### E.1  Privacy analysis – proof of Claim 6.3

Fix $t \in \mathbb{N}$ and the adversary $\mathcal{B}$. We need to show that $\text{View}^0_{\mathcal{B},t}$ and $\text{View}^1_{\mathcal{B},t}$ (defined in Figure 1) are $(\varepsilon, \delta) - indistinguishable$. We will consider separately the case where the executions differ in the training phase (Claim E.5) and the case where the difference occurs during the prediction phase (Claim E.6).

**Privacy of the initial training set** $S$.   Let $S^0, S^1 \in (X \times \{0, 1\})^n$ be neighboring datasets of labeled examples and let $\text{View}^0_{\mathcal{B},t}$ and $\text{View}^1_{\mathcal{B},t}$ be as in Figure 1 where $\left( (x_1^0, y_1^0), \ldots, (x_n^0, y_n^0) \right) = S^0$ and $\left( (x_1^1, y_1^1), \ldots, (x_n^1, y_n^1) \right) = S^1$.

**Claim E.5.** *For all adversaries $\mathcal{B}$, for all $t > 0$, and for any two neighbouring database $S^0$ and $S^1$ selected by $\mathcal{B}$, $\text{View}^0_{\mathcal{B},t}$ and $\text{View}^1_{\mathcal{B},t}$ are $(\varepsilon, \delta)$-indistinguishable.*

*Proof.* Let $R'_1 = \min(t, R_1)$. Note that $\text{View}^b_{\mathcal{B},R'_1}$ is a prefix of $\text{View}^b_{\mathcal{B},t}$ which includes the labels Algorithm `GenericBBL` produces in Step 3(d)iii for the $R'_1$ first unlabeled points selected by $\mathcal{B}$. Let $S_2^b$ be the result of the first application of algorithm `LabelBoost` in Step 3f of `GenericBBL` (if $t < R_1$ we set $S_2^b$ as $\perp$). The creation of these random variables is depicted in Figure 4, where $D_1^L$ denotes the labels Algorithm `GenericBBL` produces for the unlabeled points $D_1$.

Observe that $\text{View}^b_{\mathcal{B},t}$ results from a post-processing (jointly by the adversary $\mathcal{B}$ and Algorithm `GenericBBL`) of the random variable $\left( \text{View}^b_{\mathcal{B},R'_1}, S_2^b \right)$, and hence it suffices to show that $\left( \text{View}^0_{\mathcal{B},R'_1}, S_2^0 \right)$ and $\left( \text{View}^1_{\mathcal{B},R'_1}, S_2^1 \right)$ are $(\varepsilon, \delta)$-indistinguishable.

We follow the processes creating $\text{View}^b_{\mathcal{B},t}$ and $S_2^b$ in Figure 4: (i) The mechanism $M_1$ corresponds to the loop in Step 3d of `GenericBBL` where labels are produced for the adver-

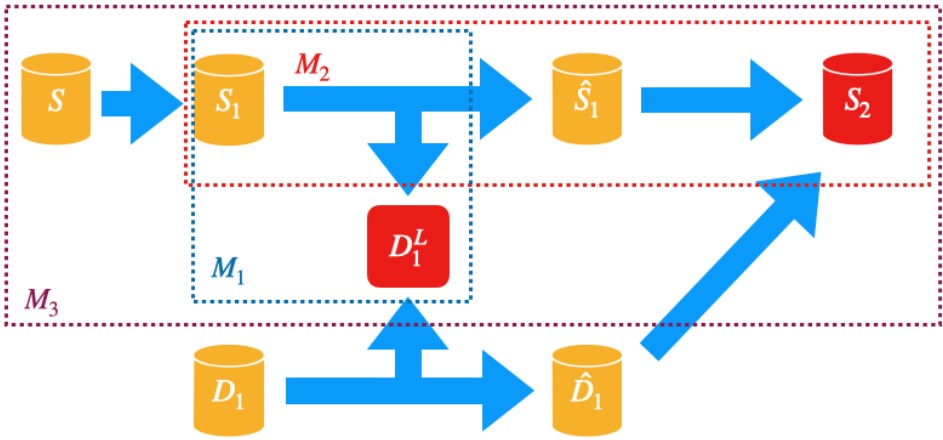

Figure 4: Privacy of the labeled sample $S$

sarially chosen points $D_1^b$. By application of claim D.1, $M_1$ is $(1,\delta)$-differentially private. (ii) The mechanism $M_2$, corresponds to the subsampling of $\hat{S}_1^b$ from $S_1^b$ and the application of procedure LabelBoost on the subsample in Step 3f of GenericBBL resulting in $S_2^b$. By application of Claim 2.9 and Lemma C.1, $M_2$ is $(\varepsilon, 0)$-differentially private. Thus $(M_1, M_2)$ is $(\varepsilon + 1, \delta)$-differentially private. (iii) The mechanism $M_3$ with input of $S^b$ and output $\left(D_1^{b,L}, S_2^b\right) = \left(\text{View}_{\mathcal{B}, R_1'}^b, S_2^b\right)$ applies $(M_1, M_2)$ on the sub-sample $S_1^b$ obtained from $S^b$ in Step 2 of GenericBBL. By application of Claim 2.9 $M_3$ is $(\varepsilon, \frac{4\varepsilon\delta}{3+\exp(\varepsilon+1)})$-differentially private. Since $\frac{4\varepsilon\delta}{3+\exp(\varepsilon+1)} \leq \delta$ for any $\varepsilon$, hence $\left(\text{View}_{\mathcal{B}, R_1'}^0, S_2^0\right)$ and $\left(\text{View}_{\mathcal{B}, R_1'}^1, S_2^1\right)$ are $(\varepsilon, \delta)$-indistinguishable $\qquad\square$

**Privacy of the unlabeled points $D$.** Let $D^0, D^1 \in X^t$ be neighboring datasets of unlabeled examples and let $\text{View}_{\mathcal{B}, t}^0$ and $\text{View}_{\mathcal{B}, t}^1$ be as in Figure 1 where $\left(x_1^0, \ldots, x_t^0\right) = D^0$ and $\left(x_1^1, \ldots, x_t^1\right) = D^1$.

**Claim E.6.** *For all adversaries $\mathcal{B}$, for all $t > 0$, and for any two neighbouring databases $D^0$ and $D^1$ selected by $\mathcal{B}$, $\text{View}_{\mathcal{B}, t}^0$ and $\text{View}_{\mathcal{B}, t}^1$ are $(\varepsilon, \delta)$-indistinguishable.*

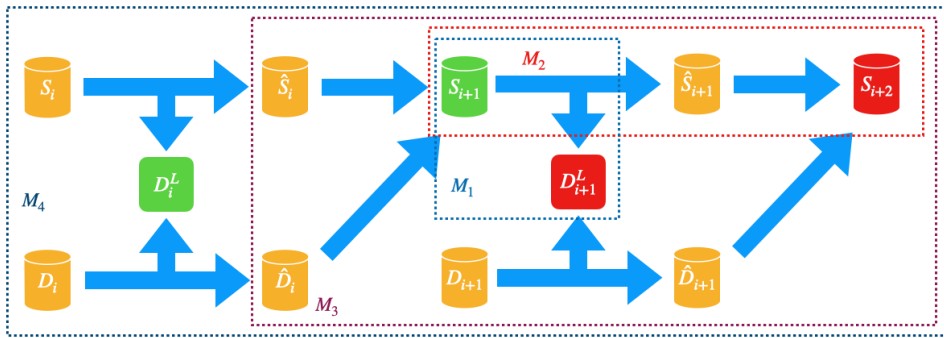

Figure 5: Privacy leakage of $D_i$

*Proof.* Let $D_1^0, D_2^0, \ldots, D_k^0$ and $D_1^1, D_2^1, \ldots, D_k^1$ be the set of unlabeled databases in step 3e of GenericBBL. Without loss of generality, we assume $D_i^0$ and $D_i^1$ differ on one entry. When

$i = k$, $\text{View}^0_{\mathcal{B},t} = \text{View}^1_{\mathcal{B},t}$ because all selected hypothesis are the same. When $i < k$, let $R' = \min\left(\sum_{j=1}^{i+1} R_j, t\right)$.

Similar to the analysis if Claim E.5, $\text{View}^b_{\mathcal{B},t}$ results from a post-processing of the random variable $(\text{View}^b_{\mathcal{B},R''}, S^b_{i+2})$ (if $t < \sum_{j=1}^{i+1} R_j$ we set $S^b_{i+2}$ as $\perp$). Note that $\text{View}^b_{\mathcal{B},R'_1} = (D_1^{b,L}, \dots, D_i^{b,L*}, D_{i+1}^{b,L})$, and $(D_1^{b,L}, \dots, D_{i-1}^{b,L}, D_i^{b,L*})$ follow the same distribution for $b \in \{0,1\}$, where $D_i^{b,L*}$ is the labels of points in $D_i^b$ expect the different point. So that it suffices to show that $\left(D_{i+1}^{0,L}, S^0_2\right)$ and $\left(D_{i+1}^{1,L}, S^1_2\right)$ are $(\varepsilon, \delta)$-indistinguishable.

We follow the processes creating $D_{i+1}^{b,L}$ and $S^b_{i+2}$ in Figure 5: (i) The mechanism $M_1$ corresponds to the loop in Step 3d of GenericBBL where labels are produced for the adversarially chosen points $D_{i+1}^b$. By application of claim D.1, $M_1$ is $(1, \delta)$-differentially private. (ii) The mechanism $M_2$, corresponds to the subsampling of $\hat{S}_{i+1}^b$ from $S_{i+1}^b$ and the application of procedure LabelBoost on the subsample in Step 3f of GenericBBL resulting in $S^b_{i+2}$. By application of Claim 2.9 and Lemma C.1, $M_2$ is $(\varepsilon, 0)$-differentially private. Thus $(M_1, M_2)$ is $(\varepsilon + 1, \delta)$-differentially private. (iii) The mechanism $M_3$ with input of $\hat{D}_i^b$ and output $\left(D_{i+1}^{b,L}, S^b_{i+2}\right)$ applies $(M_2, M_3)$ on $S_{i+1}$, which is generated from $\hat{D}_i^b$ and in Step 3f of GenericBBL. By application of Claim C.1, $M_3$ is $(\varepsilon + 4, 4\varepsilon\delta)$-differentially private. (iv) The mechanism $M_4$, corresponds to the subsampling $\hat{D}_i^b$ from $D_i^b$ and the application of $M_4$ on $\hat{D}_i^b$. By application of Claim 2.9, $M_4$ is $(\varepsilon, \frac{16e\varepsilon\delta}{3+\exp(\varepsilon+4)})$-differentially private. Since $\frac{16e\varepsilon}{3+\exp(\varepsilon+4)} \leq 1$ for any $\varepsilon$, $\left(D_{i+1}^{0,L}, S^0_2\right)$ and $\left(D_{i+1}^{1,L}, S^1_2\right)$ are $(\varepsilon, \delta)$-indistinguishable. $\qquad\square$

**Remark E.7.** *The above proofs work on the adversarially selected D because: (i) Lemma ?? works on the adaptively selected queries. (We treat the hypothesis class $F_i$ as the database, the unlabelled points $x_{i,\ell}$ as the query parameters.) (ii) LabelBoost generates labels by applying one private hypothesis on points. The labels are differentially private by post-processing.*

