# Private Everlasting Prediction

## Abstract

A private learner is trained on a sample of labeled points and generates a hypothesis that can be used for predicting the labels of newly sampled points while protecting the privacy of the training set [Kasiviswannathan et al., FOCS 2008]. Research uncovered that private learners may need to exhibit significantly higher sample complexity than non-private learners as is the case with, e.g., learning of one-dimensional threshold functions [Bun et al., FOCS 2015, Alon et al., STOC 2019].

We explore prediction as an alternative to learning. Instead of putting forward a hypothesis, a predictor answers a stream of classification queries. Earlier work has considered a private prediction model with just a single classification query [Dwork and Feldman, COLT 2018]. We observe that when answering a stream of queries, a predictor must modify the hypothesis it uses over time, and, furthermore, that it must use the queries for this modification, hence introducing potential privacy risks with respect to the queries themselves.

We introduce *private everlasting prediction* taking into account the privacy of both the training set *and* the (adaptively chosen) queries made to the predictor. We then present a generic construction of private everlasting predictors in the PAC model. The sample complexity of the initial training sample in our construction is quadratic (up to polylog factors) in the VC dimension of the concept class. Our construction allows prediction for all concept classes with finite VC dimension, and in particular threshold functions with constant size initial training sample, even when considered over infinite domains, whereas it is known that the sample complexity of privately learning threshold functions must grow as a function of the domain size and hence is impossible for infinite domains.

## 1  Introduction

A PAC learner is given labeled examples $S = \{(x_i, y_i)\}_{i \in [n]}$ drawn i.i.d. from an unknown underlying probability distribution $\mathcal{D}$ over a data domain $X$ and outputs a hypothesis $h$ that can be used for predicting the label of fresh points $x_{n+1}, x_{n+2}, \ldots$ sampled from the same underlying probability distribution $\mathcal{D}$ [Valiant, 1984]. It is well known that when points are labeled by a concept selected from a concept class $C = \{c : X \rightarrow \{0, 1\}\}$ then learning is possible with sample complexity proportional to the VC dimension of the concept class.

Learning often happens in settings where the underlying training data is related to individuals and privacy-sensitive and where a learner is required, for legal, ethical, or other reasons, to protect personal information from being leaked in the learned hypothesis $h$. Private learning was introduced by Kasiviswanathan et al. [2011], as a theoretical model for

Submitted to 37th Conference on Neural Information Processing Systems (NeurIPS 2023). Do not distribute.

studying such tasks. A *private learner* is a PAC learner that preserves differential privacy with respect to its training set $S$. That is, the learner's distribution on outcome hypotheses must not depend too strongly on any single example in $S$. Kasiviswanathan et al. showed via a generic construction that any finite concept class can be learned privately and with sample complexity $n = O(\log|C|)$. This value ($O(\log|C|)$) can be significantly higher than the VC dimension of the concept class $C$ (see below).

It is now understood that the gap between the sample complexity of private and non-private learners is essential – an important example is private learning of threshold functions (defined over an ordered domain $X$ as $C_{thresh} = \{c_t\}_{t \in X}$ where $c_t(x) = \mathbb{1}_{x \geq t}$), which requires sample complexity that is asymptotically higher than the (constant) VC dimension of $C_{thresh}$. In more detail, with *pure* differential privacy, the sample complexity of private learning is characterized by the representation dimension of the concept class [Beimel et al., 2013a]. The representation dimension of $C_{thresh}$ (hence, the sample complexity of private learning thresholds) is $\Theta(\log|X|)$ [Feldman and Xiao, 2015]. With *approximate* differential privacy, the sample complexity of learning threshold functions is $\Theta(\log^*|X|)$ [Beimel et al., 2013b, Bun et al., 2015, Alon et al., 2019, Kaplan et al., 2020, Cohen et al., 2022]. Hence, in both the pure and approximate differential privacy cases, the sample complexity grows with the cardinality of the domain $|X|$ and no private learner exists for threshold functions over infinite domains, such as the integers and the reals, whereas low sample complexity non-private learners exist for these tasks.

**Privacy preserving (black-box) prediction.** Dwork and Feldman [2018] proposed privacy-preserving prediction as an alternative for private learning. Noting that "[i]t is now known that for some basic learning problems [. . .] producing an accurate private model requires much more data than learning without privacy," they considered a setting where "users may be allowed to query the prediction model on their inputs only through an appropriate interface". That is, a setting where the learned hypothesis is not made public. Instead, it may be accessed in a "black-box" manner via a privacy-preserving query-answering prediction interface. The prediction interface is required to preserve the privacy of its training set $S$:

**Definition 1.1** (private prediction interface [Dwork and Feldman, 2018] (rephrased)). A prediction interface $M$ is $(\epsilon, \delta)$-differentially private if for every interactive query generating algorithm $Q$, the output of the interaction between $Q$ and $M(S)$ is $(\epsilon, \delta)$-differentially private with respect to $S$.

Dwork and Feldman focused on the setting where the entire interaction between $Q$ and $M(S)$ consists of issuing a single prediction query and answering it:

**Definition 1.2** (Single query prediction [Dwork and Feldman, 2018]). Let $M$ be an algorithm that given a set of labeled examples $S$ and an unlabeled point $x$ produces a label $y$. $M$ is an $(\epsilon, \delta)$-differentially private prediction algorithm if for every $x$, the output $M(S, x)$ is $(\epsilon, \delta)$-differentially private with respect to $S$.

W.r.t. answering a single prediction query, Dwork and Feldman showed that the sample complexity of such predictors is proportional to the VC dimension of the concept class.

## 1.1 Our contributions

In this work, we extend private prediction beyond a single query to answering any sequence – *unlimited in length* – of prediction queries. We refer to this as *private everlasting prediction*. Our goal is to present a generic private everlasting predictor with low training sample complexity $|S|$.

**Private prediction interfaces when applied to a large number of queries.** We begin by examining private everlasting prediction under the framework of Definition 1.1. We prove:

**Theorem 1.3** (informal version of Theorem 3.3). *Let $\mathcal{A}$ be a private everlasting prediction interface for concept class $C$ and assume $\mathcal{A}$ bases its predictions solely on the initial training set $S$, then there exists a private learner for concept class $C$ with sample complexity $|S|$.*

This means that everlasting predictors that base their prediction solely on the initial training set $S$ are subject to the same complexity lowerbounds as private learners. Hence, to avoid

private learning lowerbounds, private everlasting predictors need to rely on more than the initial training sample $S$ as a source of information about the underlying probability distribution and the labeling concept.

In this work, we choose to allow the everlasting predictor to rely on the queries made - which are unlabeled points from the domain $X$, assuming the queries are drawn from the same distribution the initial training $S$ is sampled from. This requires changing the privacy definition, as Definition 1.1 does not protect the queries made, yet the classification given to a query can now depend on and hence reveal information provided in queries made earlier.

**A definition of private everlasting predictors.** Our definition of private everlasting predictors is motivated by the observations above. Consider an algorithm $\mathcal{A}$ that is first fed with a training set $S$ of labeled points and then executes for an unlimited number of rounds, where in round $i$ algorithm $\mathcal{A}$ receives as input a query point $x_i$ and produces a label $\hat{y}_i$. We say that $\mathcal{A}$ is an everlasting predictor if, when the (labeled) training set $S$ and the (unlabeled) query points are coming from the same underlying distribution, $\mathcal{A}$ answers each query points $x_i$ with a good hypothesis $h_i$, and hence the label $\hat{y}_i$ produced by $\mathcal{A}$ is correct with high probability. We say that $\mathcal{A}$ is a *private* everlasting predictor if its sequence of predictions $\hat{y}_1, \hat{y}_2, \hat{y}_3, \ldots$ protects both the privacy of the training set $S$ *and* the query points $x_1, x_2, x_3, \ldots$ in face of any adversary that adaptively chooses the query points.

We emphasize that while private everlasting predictors need to exhibit average-case utility – as good prediction is required only for the case where $S$ and $x_1, x_2, x_3, \ldots$ are selected i.i.d. from the same underlying distribution – our privacy requirement is worst-case, and holds in face of an *adaptive* adversary that chooses each query point $x_i$ after receiving the prediction provided for $(x_1, \ldots, x_{i-1})$, and not necessarily in accordance with any probability distribution.

**A generic construction of private everlasting predictors.** Our construction, called `GenericBBL`, executes in rounds. The input to the first round is the initial labeled training set $S$, where the number of samples in $S$ is quadratic in the VC dimension of the concept class. Each other round begins with a collection $S_i$ of labeled examples and ends with newly generated collection of labeled examples $S_{i+1}$. The set $S$ is assumed to be consistent with some concept $c \in C$ and our construction ensures that this is the case also for the sets $S_i$ for all $i$. We briefly describe the main computations performed in each round of `GenericBBL`.[1]

- **Round initialization:** At the outset of a round, the labeled set $S_i$ is partitioned into sub-sets, each with number of samples which is proportional to the VC dimension (so we have $\approx \frac{|S_i|}{\mathrm{VC}(C)}$ sub-sets). Each of the sub-sets is used for training a classifier non-privately, hence creating a collection of classifiers $F_i = \{f : X \to \{0, 1\}\}$ that are used throughout the round.

- **Query answering:** Queries are issued to the predictor in an online manner. Each query is first labeled by each of the classifiers in $F_i$. Then the predicted label is computed by applying a privacy-preserving majority vote on these intermediate labels. (By standard composition theorems for differential privacy, we could answer roughly $|F_i|^2 \approx \left(\frac{|S_i|}{\mathrm{VC}(C)}\right)^2$ queries without exhausting our privacy budget.) To save on the privacy budget, the majority vote is based on the `BetweenThresholds` mechanism of Bun et al. [2016] (which in turn is based on the sparse vector technique). The algorithm fails when the privacy budget is exhausted. However, when queries are sampled from the underlying distribution then with a high enough probability the labels produced by the classifiers in $F_i$ would exhibit a clear majority.

- **Generating a labeled set for the following round:** The predictions provided in the duration of a round are not guaranteed to be consistent with any concept in $C$ and hence cannot be used to set the following round. Instead, at the end of the round these points are relabeled consistently with $C$ using a technique developed by Beimel et al.

---

[1]Important details, such as privacy amplification via sampling and management of the learning accuracy and error parameters are omitted from the description provided in this section.

[2021] in the context of private semi-supervised learning. Let $S_{i+1}$ denote the query points obtained during the $i$th round, after (re)labeling them. This is a collection of size $|S_{i+1}| \approx \left(\frac{|S_i|}{\text{VC}(C)}\right)^2$. Hence, provided that $|S_i| \gtrsim (\text{VC}(C))^2$ we get that $|S_{i+1}| > |S_i|$ which allows us to continue to the next round with more data than we had in the previous round.

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

**Algorithm `BetweenThresholds` [Bun et al., 2016]:** Algorithm `BetweenThresholds` takes as input a database $S \in X^n$ and thredholds $t_\ell, t_u$. It applies the sparse vector technique to answer noisy threshold queries with $L$ (below threshold) $R$ (above threshold) and $\top$ (halt). We use the following lemmata by Bun et al. [2016] and observe that, using standard privacy amplification theorems, Algorithm `BetweenThresholds` can be modified to allow for $c$ times of outputting $\top$ before halting, with a (roughly) $\sqrt{c}$ growth in its privacy parameter.

**Lemma 4.3** (Privacy for `BetweenThresholds`). *Let $\varepsilon, \delta \in (0, 1)$ and $n \in \mathbb{N}$. Then algorithm `BetweenThresholds` is $(\varepsilon, \delta)$-differentially private for any adaptively-chosen sequence of queries as long as the gap between the thresholds $t_\ell, t_u$ satisfies $t_u - t_\ell \geq \frac{12}{\varepsilon n} (\log(10/\varepsilon) + \log(1/\delta) + 1)$.*

**Lemma 4.4** (Accuracy of `BetweenThresholds`). *Let $\alpha, \beta, \varepsilon, t_\ell, t_u \in (0, 1)$ and $n, k \in \mathbb{N}$ satisfy $n \geq \frac{8}{\alpha \varepsilon} (\log(k + 1) + \log(1/\beta))$. Then, for any input $x \in X^n$ and any adaptively-chosen sequence of queries $q_1, q_2, \cdots, q_k$, the answers $a_1, a_2, \cdots a_{\leq k}$ produced by `BetweenThresholds` on input $x$ satisfy the following with probability at least $1 - \beta$. For any $j \in [k]$ such that $a_j$ is returned before `BetweenThresholds` halts, (i) $a_j = \text{L} \implies q_j(x) \leq t_\ell + \alpha$, (ii) $a_j = \text{R} \implies q_j(x) \geq t_u - \alpha$, and (iii) $a_j = \top \implies t_\ell - \alpha \leq q_j(x) \leq t_u + \alpha$.*

**Observation 1.** *Using standard composition theorems for differential privacy (see, e.g., Dwork et al. [2010]), we can assume that algorithm `BetweenThresholds` takes another parameter $c$, and halts after $c$ times of outputting $\top$. In this case, the algorithm satisfies $(\varepsilon', 2c\delta)$-differential privacy, for $\varepsilon' = \sqrt{2c \ln(\frac{1}{c\delta})} \varepsilon + c\varepsilon(e^\varepsilon - 1)$.*

## 5  A Generic Construction

Our generic construction Algorithm `GenericBBL` transforms a (non-private) learner for a concept class $C$ into a private everlasting predictor for $C$. The proof of the following theorem follows from Theorem 5.2 and Claim 5.3 which are proved in Appendix E.

**Theorem 5.1.** *Given $\alpha, \beta, \delta < 1/16, \epsilon < 1$, Algorithm `GenericBBL` is a $(6\alpha, 4\beta, \epsilon, \delta, n)$-private everlasting predictor, where $n$ is set as in Algorithm `GenericBBL`.*

---

**Algorithm `GenericBBL`**

---

**Initial input:** A labeled database $S \in (X \times \{0,1\})^n$ where $n = \frac{8\tau}{\alpha^3 \epsilon^2} \cdot$ $\left(8\mathrm{VC}(C)\log(\frac{26}{\alpha}) + 4\log(\frac{4}{\beta})\right)^2 \cdot \log(\frac{1}{\delta}) \cdot \log^2\left(\frac{64\mathrm{VC}(C)\log(\frac{26}{\alpha}) + 32\log(\frac{4}{\beta})}{\epsilon\alpha^2\beta\delta}\right) \cdot (3 + \exp(\epsilon + 4))$.

1. Let $\tau > 1.1 * 10^{10}$. Set $\alpha_1 = \alpha/2, \beta_1 = \beta/2$. Define $\lambda_i = \frac{8\mathrm{VC}(C)\log(\frac{13}{\alpha_i}) + 4\log(\frac{2}{\beta_i})}{\alpha_i}$.
   /* by Theorem A.2 $\lambda_i$ samples suffice for PAC learning $C$ with parameters $\alpha_i, \beta_i$ */

2. Let $S_1 \subseteq S$ be a random subset of size $n \cdot \frac{\epsilon}{3+\exp(\epsilon+4)} = \frac{\tau \cdot \lambda_i^2 \cdot \log(\frac{1}{\delta}) \cdot \log^2(\frac{\lambda_i}{\epsilon\alpha_i\beta_i\delta})}{\alpha_i\epsilon}$.

3. Repeat for $i = 1, 2, 3, \ldots$

   (a) Divide $S_i$ into $T_i = \frac{\tau \cdot \lambda_i \cdot \log(\frac{1}{\delta}) \cdot \log^2(\frac{\lambda_i}{\epsilon\alpha_i\beta_i\delta})}{\alpha_i\epsilon}$ disjoint databases $S_{i,1}, \ldots, S_{i,T_i}$ of size $\lambda_i$.

   (b) For $t \in [T_i]$ let $f_t \in C$ be a hypothesis minimizing $\mathrm{error}_{S_{i,t}}(\cdot)$. Define $F_i = (f_1, \ldots, f_{T_i})$.

   (c) Set $R_i = \frac{25600|S_i|}{\epsilon}$. Set $t_u = 1/2 + \alpha_i, t_\ell = 1/2 - \alpha_i$. Set the privacy parameters $\epsilon_i' = \frac{1}{3\sqrt{c_i \ln(\frac{2}{\delta})}}$ and $\delta_i' = \frac{\delta}{2c_i}$, where $c_i = 64\alpha_i R_i$. Instantiate algorithm `BetweenThresholds` on the database of hypotheses $F_i$ allowing for $c_i = 64\alpha_i R_i$ rounds of $\top$ while satisfying $(1, \delta)$-differential privacy (as in Observation 2).

   (d) For $\ell = 1$ to $R_i$:
      i. Receive as input a prediction query $x_{i,\ell} \in X$.
      ii. Give `BetweenThresholds` the query $q_{x_{i,\ell}}$ where $q_{x_{i,\ell}}(F_i) = \sum_{t \in [T_i]} f_t(x_{i,\ell})$, and obtain an outcome $y_{i,\ell} \in \{L, \top, R\}$.
      iii. Respond with the label 0 if $y_{i,\ell} = L$ and 1 if $y_{i,\ell} \in \{R, \top\}$.
      iv. If `BetweenThresholds` halts, then halt and fail (recall that `BetweenThresholds` only halts if $c_i$ copies of $\top$ were encountered during the current iteration).

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

## C.2 Algorithm `BetweenThresholds` [Bun et al., 2016]

---

**Algorithm `BetweenThresholds` [Bun et al., 2016]**

---

**Input:** Database $S \in X^n$.
**Parameters:** $\varepsilon, t_\ell, t_u \in (0, 1)$ and $n, k \in \mathbb{N}$.
    1. Sample $\mu \sim \text{Lap}(2/\varepsilon n)$ and initialize noisy thresholds $\hat{t}_\ell = t_\ell + \mu$ and $\hat{t}_u = t_u - \mu$.
    2. For $j = 1, 2, \cdots, k$:
        (a) Receive query $q_j : X^n \to [0, 1]$.
        (b) Set $c_j = q_j(S) + \nu_j$ where $\nu_j \sim \text{Lap}(6/\varepsilon n)$.
        (c) If $c_j < \hat{t}_\ell$, output L and continue.
        (d) If $c_j > \hat{t}_u$, output R and continue.
        (e) If $c_j \in [\hat{t}_\ell, \hat{t}_u]$, output $\top$ and halt.

---

**Lemma C.3** (Privacy for `BetweenThresholds` [Bun et al., 2016]). *Let $\varepsilon, \delta \in (0, 1)$ and $n \in \mathbb{N}$. Then algorithm `BetweenThresholds` is $(\varepsilon, \delta)$-differentially private for any adaptively-chosen sequence of queries as long as the gap between the thresholds $t_\ell, t_u$ satisfies*

$$t_u - t_\ell \ge \frac{12}{\varepsilon n}\left(\log(10/\varepsilon) + \log(1/\delta) + 1\right).$$

**Lemma C.4** (Accuracy for `BetweenThresholds` [Bun et al., 2016]). *Let $\alpha, \beta, \varepsilon, t_\ell, t_u \in (0, 1)$ and $n, k \in \mathbb{N}$ satisfy*

$$n \ge \frac{8}{\alpha\varepsilon}\left(\log(k + 1) + \log(1/\beta)\right).$$

*Then, for any input $x \in X^n$ and any adaptively-chosen sequence of queries $q_1, q_2, \cdots, q_k$, the answers $a_1, a_2, \cdots a_{\le k}$ produced by `BetweenThresholds` on input $x$ satisfy the following with probability at least $1 - \beta$. For any $j \in [k]$ such that $a_j$ is returned before `BetweenThresholds` halts,*

- $a_j = \text{L} \implies q_j(x) \le t_\ell + \alpha$,

- $a_j = \text{R} \implies q_j(x) \ge t_u - \alpha$, *and*

- $a_j = \top \implies t_\ell - \alpha \le q_j(x) \le t_u + \alpha$.

**Observation 2.** *Using standard composition theorems for differential privacy (see, e.g., Dwork et al. [2010]), we can assume that algorithm `BetweenThresholds` takes another parameter $c$, and halts after $c$ times of outputting $\top$. In this case, the algorithm satisfies $(\varepsilon', 2c\delta)$-differential privacy, for $\varepsilon' = \sqrt{2c\ln(\frac{1}{c\delta})}\varepsilon + c\varepsilon(e^\varepsilon - 1)$.*

## D  Some Technical Facts

We refer to the execution of steps 3a-3g of algorithm GenericBBL as a *phase* of the algorithm, indexed by $i = 1, 2, 3, \ldots$.

The original BetweenThresholds needs to halt when it outputs $\top$. In GenericBBL, we tolerance it to halt at most $c_i$ times in the phase $i$. We prove BetweenThresholds in GenericBBL is $(1, \delta)$-differentially private.

**Claim D.1.** *For $\delta < 1$, Mechanism* BetweenThresholds *used in step 3c in the i-th iteration, is* $(1, \delta)$*-differentially private.*

*Proof.* Let $\varepsilon_i', \delta_i'$ be as in Step 3c. Since $e^{\varepsilon_i'} - 1 < 2\varepsilon_i'$ for $0 < \varepsilon_i' < 1$, we have

$$\sqrt{2c_i \ln\left(\frac{1}{c_i \delta_i'}\right)} \cdot \varepsilon_i' + c_i \varepsilon_i'(e^{\varepsilon_i'} - 1) \leq \sqrt{2c_i \ln\left(\frac{2}{\delta}\right)} \cdot \varepsilon_i' + 2c_i \varepsilon_i'^2 = \frac{\sqrt{2}}{3} + \frac{2}{9 \ln(\frac{2}{\delta})} \leq 1.$$

The proof is concluded by using observation 2. $\qquad\qquad\square$

In Claim D.2- D.5, we prove that with high probability, BetweenThresholds in step 3d halts within $64\alpha_i$ times. We prove it by 4 steps:
1.prove that with high probability, most hypothesis in step 3b have high accuracy (Claim D.2).
2.prove that if most hypothesis in step 3b have high accuracy, then with high probability, the queries in BetweenThresholds are closed to 0 or 1 (Claim D.3).
3.prove that if the queries in BetweenThresholds are closed to 0 or 1, then BetweenThresholds in step 3d will outputs $L$ or $R$ with high probability(Claim D.4).
4.prove that if BetweenThresholds outputs $L$ or $R$, then every single phase fails with low probability(Claim D.5).

**Claim D.2.** *If $\beta_i \leq 1/32$ and $T_i \geq 96 \ln \frac{1}{\alpha_i}$, then with probability $1 - \alpha_i$, $\frac{15T_i}{16}$ hypotheses in step 3b are $\alpha_i$-good with respect to $g_i$, where $g_i$ is the concept of $S_i$.*

*Proof.* By the VC bound (Theorem A.2), for each $t \in [T_i]$, we have

$$\Pr[\text{error}_{\mathcal{D}}(f_t, g_i) \leq \alpha_i] \geq 1 - \beta_i.$$

By Chernoff bound, if $T_i \geq \frac{16 + 256\beta_i}{(1 - 16\beta_i)^2} \ln \frac{1}{\alpha_i}$, then with probability $1 - \alpha_i$, we have $\frac{15T_i}{16}$ hypotheses have $\text{error}_{\mathcal{D}}(f_t, g_i) \leq \alpha_i$. When $\beta_i \leq 1/32$, it is sufficient to set $T_i \geq 96 \ln \frac{1}{\alpha_i}$. $\qquad\square$

**Claim D.3.** *If $\alpha_i \leq 1/16$ and $\frac{15T_i}{16}$ hypotheses in step 3b are $\alpha_i$-good with respect to $g_i$, where $g_i$ is the concept of $S_i$, then $\Pr_{x \sim \mathcal{D}}[|q(x) - \frac{1}{2}| \leq \frac{3}{8}] \leq 15\alpha_i$.*

*Proof.* W.l.o.g. assume $g_i(x) = 1$, where $g_i$ is the concept of $S_i$, so it is sufficient to prove $\Pr_{x \sim \mathcal{D}}[q(x) \leq \frac{7}{8}] \leq 8\alpha_i$. Consider the worst case that $\frac{T_i}{16}$ "bad" hypotheses output 0. In that case, $q(x) \leq \frac{7}{8}$ when $\frac{T_i}{16}$ of $\alpha_i$-good hypotheses output 0. So that with probability $15\alpha_i$, we have $q(x) \leq \frac{7}{8}$.(see Figure 2)

$\qquad\qquad\square$

**Claim D.4.** *Let $t_u < 1/2 + 1/8$ and $t_\ell > 1/2 - 1/8$. For a query $q$ such that $q(S) > 7/8$ (similarly, for $q(S) < 1/8$), Algorithm* BetweenThresholds *outputs $R$ (similarly, $L$) with probability at least*

$$1 - \exp\left(-\frac{T_i}{144\sqrt{c_i \ln(\frac{2}{\delta})}}\right).$$

*Proof.* Wlog assume $q(S) > 7/8$, it is sufficient to show

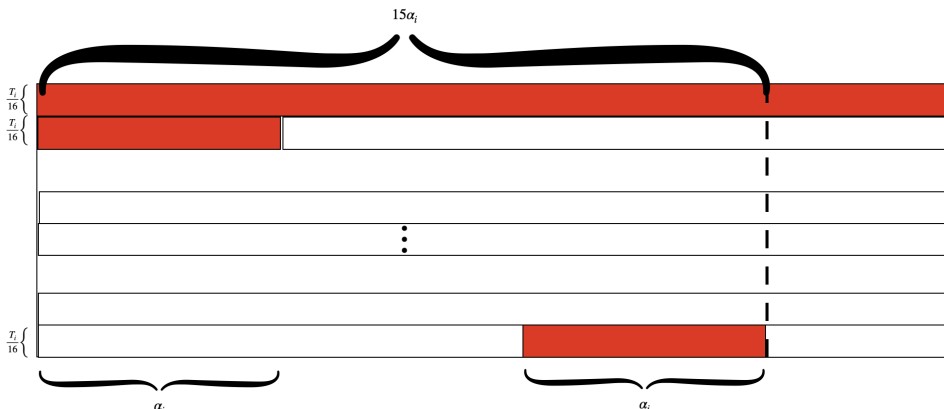

Figure 2: The horizontal represents the input point. The vertical represents the hypothesis. The red parts represent the incorrect prediction. We let $\frac{T_i}{16}$ hypothesis predict all labels as 0. To let $q(x) \le \frac{7}{8}$, there must exist $\frac{T_i}{16}$ hypothesis output 0. In the worst case, at most $15\alpha_i$ of points are labeled as 0.

$$
\begin{aligned}
\Pr[\texttt{BetweenThreshold outputs} R] &= \Pr[q(S) + \mathrm{Lap}(6/\varepsilon' T_i) > t_u + \mathrm{Lap}(2/\varepsilon' T_i)] \\
&> \Pr[\mathrm{Lap}(6/\varepsilon' T_i) > -1/8] \cdot \Pr[\mathrm{Lap}(2/\varepsilon' T_i) < 1/8] \\
&= \left( 1 - \frac{1}{2}\exp\left( -\frac{T_i}{144\sqrt{c_i \ln(\frac{2}{\delta})}} \right) \right) \cdot \left( 1 - \frac{1}{2}\exp\left( -\frac{T_i}{48\sqrt{c_i \ln(\frac{2}{\delta})}} \right) \right) \\
&> 1 - \exp\left( -\frac{T_i}{144\sqrt{c_i \ln(\frac{2}{\delta})}} \right).
\end{aligned}
$$

$\square$

**Claim D.5.** *For any phase $i$, `BetweenThresholds` outputs $\top$ at most $64\alpha_i R_i$ times with probability at most $\beta_i$.*

*Proof.* For a single query, if $t_u < 1/2 + 1/8$ and $q(S) > 7/8$ (similarly, $t_\ell > 1/2 - 1/8$ and $q(S) < 1/8$), by Claim D.4, `BetweenThresholds` outputs $\top$ with probability at most $\exp\left( -\frac{T_i}{144\sqrt{c_i \ln(\frac{2}{\delta})}} \right) = \exp\left( -\frac{T_i}{144\sqrt{64\alpha_i R_i \ln(\frac{2}{\delta})}} \right) < \alpha_i$. Combine Claim D.2 and D.3, `BetweenThresholds` outputs $\top$ with probability at most $32\alpha_i$. By the Chernoff bound and $R_i \ge \frac{3\ln(\frac{1}{\beta_i})}{\alpha_i}$, `BetweenThresholds` outputs $\top$ more than $64\alpha_i R_i$ times with probability at most $\beta_i$. $\square$

In step 3f, `GenericBBL` takes a random subset of size $\lambda_{i+1} T_{t+1}$ from $\hat{D}'_i$. We show that the size of $\hat{D}'_i$ is at least $\lambda_{i+1} T_{t+1}$.

**Claim D.6.** *When $\varepsilon \le 1$, for any $i \ge 1$, we always have $|\hat{D}'_i| \ge \lambda_{i+1} T_{i+1}$.*

*Proof.* Let $m = 3 + \exp(\varepsilon + 4) < 200$. By the step 3c, step 3e and step 3f, $|\hat{D}_j| = \frac{\varepsilon |D_j|}{m} = \frac{25600|S_j|}{m} \ge 128|S_j| = 128\lambda_j T_j$. Then it is sufficient to verify $128\lambda_j T_j \ge \lambda_{j+1} T_{j+1}$

We can verify that

$$4\lambda_j = 4 \cdot \frac{8\text{VC}(C)\log(\frac{13}{\alpha_i}) + 4\log(\frac{2}{\beta_i})}{\alpha_i} = 4 \cdot \frac{8\text{VC}(C)(\log(\frac{13}{\alpha_{j+1}}) - 1) + 4(\log(\frac{2}{\beta_{j+1}}) - 1)}{2\alpha_{j+1}} \geq \lambda_{j+1}$$

and

$$32T_j = \frac{32\tau \cdot \lambda_i \cdot \log(\frac{1}{\delta}) \cdot \log^2(\frac{\lambda_i}{\varepsilon\alpha_i\beta_i\delta})}{\alpha_i\varepsilon} \geq \frac{32\tau \cdot \lambda_i \cdot \log(\frac{1}{\delta}) \cdot \log^2(\frac{\lambda_{i+1}}{16\varepsilon\alpha_{i+1}\beta_{i+1}\delta})}{8\alpha_{i+1}\varepsilon} \geq \lambda_{j+1}T_{j+1}.$$

The last inequalitu holds because $\lambda_j \geq 4$ and $\alpha_j, \beta_j \leq 1/2$. $\qquad\square$

To apply the privacy and accuracy of $LabelBoost$ and $BetweenThresholds$, the sizes of the databases need to satisfy the inequalities in lemma C.2, C.3 and C.4. We verify that in each phase, the sizes of the databases always satisfy the requirement.

**Claim D.7.** *Let* $\alpha, \beta, \delta < 1/16$, $\varepsilon \leq 1$, *and* $\text{VC}(C) \geq 1$. *Then for any* $i \geq 1$, *we have*

$$T_i \geq \frac{8}{\alpha_i\varepsilon'}(\log(|D_i| + 1) + \log(1/\beta_i)).$$

*Proof.* By claim D.6 and step 3c, $|D_i| = \frac{25600|S_i|}{\varepsilon} = \frac{25600\lambda_iT_i}{\varepsilon}$. Since

$$\frac{8}{\alpha_i\varepsilon'}(\log(|D_i| + 1) + \log(1/\beta_i)) = \frac{24\sqrt{64\alpha_i|D_i|\ln(\frac{2}{\delta})}}{\sqrt{2}\alpha_i} \cdot (\log(|D_i| + 1) + \log(1/\beta_i))$$

$$= O\left(\sqrt{\frac{\lambda_iT_i\log(\frac{1}{\delta})}{\alpha_i\varepsilon}}\left(\log(\frac{\lambda_iT_i}{\varepsilon\beta_i})\right)\right)$$

$$= O\left(\sqrt{\frac{\lambda_iT_i\log(\frac{1}{\delta})}{\alpha_i\varepsilon}} \cdot \log\left(\frac{\lambda_i\log(\frac{1}{\delta})}{\alpha_i\beta_i\varepsilon}\right)\right),$$

and $T_i = \frac{\tau \cdot \lambda_i \cdot \log(\frac{1}{\delta}) \cdot \log^2(\frac{\lambda_i}{\varepsilon\alpha_i\beta_i\delta})}{\alpha_i\varepsilon}$, where $\tau \geq 1.1 * 10^{10}$, the inequality always holds. $\qquad\square$

**Claim D.8.** *When* $\varepsilon \leq 1$, *for any* $i \geq 1$, *we have* $|\hat{D}_i| \leq \frac{\beta_i}{e}\text{VC}(C)exp\left(\frac{\alpha_i|\hat{S}_i|}{2\text{VC}(C)}\right) - |\hat{S}_i|$.

*Proof.* By claim D.6, step 3c and step 3f,

$$|\hat{D}_i| = \frac{\varepsilon|D_i|}{m} = O(\lambda_iT_i) = O\left(\text{VC}(C)\log^2(\text{VC}(C)) \cdot \text{poly}\left(\frac{1}{\alpha_i}, \log(\frac{1}{\beta_i}), \frac{1}{\varepsilon}, \log(\frac{1}{\delta})\right)\right)$$

and

$$|\hat{S}_i| = \frac{\varepsilon|S_i|}{m}$$
$$= O(\varepsilon\lambda_iT_i) = O(\lambda_iT_i) \tag{2}$$
$$= O\left(\text{VC}(C)\log^2(\text{VC}(C)) \cdot \text{poly}\left(\frac{1}{\alpha_i}, \log(\frac{1}{\beta_i}), \frac{1}{\varepsilon}, \log(\frac{1}{\delta})\right)\right).$$

Note that

$$\frac{\beta_i}{e}\text{VC}(C)exp\left(\frac{\alpha_i|\hat{S}_i|}{2\text{VC}(C)}\right) = \Omega\left(\text{VC}^2(C) \cdot \exp\left(\text{poly}\left(\frac{1}{\alpha_i}, \log(\frac{1}{\beta_i}), \frac{1}{\varepsilon}, \log(\frac{1}{\delta})\right)\right)\right),$$

for $T_i = \frac{\tau \cdot \lambda_i \cdot \log(\frac{1}{\delta}) \cdot \log^2(\frac{\lambda_i}{\varepsilon\alpha_i\beta_i\delta})}{\alpha_i\varepsilon}$, the inequality holds when $\tau \geq 1$. $\qquad\square$

**Claim D.9.** *For every $i \geq 1$, we have*

$$t_u - t_\ell \geq \frac{12}{\varepsilon_i' T_i}\left(\log(10/\varepsilon_i') + \log(1/\delta_i') + 1\right).$$

*Proof.* By step 3c, $t_u - t_\ell = 2\alpha_i$. Then we have

$$
\begin{aligned}
\frac{6}{\alpha_i \varepsilon_i' T_i}\left(\log(10/\varepsilon_i') + \log(1/\delta_i') + 1\right) &= \frac{6\sqrt{64\alpha_i R_i \ln(\frac{2}{\delta})}}{\alpha_i T_i}\left(\log(10/\varepsilon_i') + \log(1/\delta_i') + 1\right) \\
&= 6\sqrt{\frac{1638400\ln(\frac{2}{\delta})\lambda_i}{\alpha_i T_i}}\left(\log(10/\varepsilon_i') + \log(1/\delta_i') + 1\right) \\
&= 6\sqrt{\frac{1638400\ln(\frac{2}{\delta})}{\tau \log(\frac{1}{\delta})\log^2(\frac{\lambda_i}{\varepsilon\alpha_i\beta_i\delta})}}\left(\log(10/\varepsilon_i') + \log(1/\delta_i') + 1\right) \\
&= O(1),
\end{aligned}
$$

the inequality holds when $\tau > 10^{10}$. $\square$

# E   Accuracy of Algorithm `GenericBBL` – proof of Theorem 5.2

We refer to the execution of steps 3a-3g of algorithm `GenericBBL` as a *phase* of the algorithm, indexed by $i = 1, 2, 3, \ldots$.

We give some technical facts in Appendix D. In Claim E.1, we show that in each phase, samples are labeled with high accuracy. In Claim E.2, we prove that algorithm `GenericBBL` fails with low probability. In Claim E.4, we prove that algorithm `GenericBBL` predict the labels with high accuracy.

**Claim E.1.** *When Algorithm `GenericBBL` does not fail on phases 1 to $i$, then for phase $i + 1$ we have*

$$\Pr\left[\exists g_{i+1} \in C \text{ s.t. } \text{error}_{S_{i+1}}(g_{i+1}) = 0 \text{ and } \text{error}_{\mathcal{D}}(g_{i+1}, c) \leq \sum_{j=1}^{i+1} \alpha_j\right] \geq 1 - 2\sum_{j=0}^{i+1} \beta_j.$$

*Proof.* The proof is by induction on $i$. The base case for $i = 1$ is trivial, with $g_1 = c$. Assume the claim holds for all $j \leq i$. By the properties of `LabelBoost` (Lemma C.2) and Claim D.8, with probability at least $1 - \beta_{i+1}$ we have that $S_{i+1}$ is labeled by a hypothesis $g_{i+1} \in C$ s.t. $\text{error}_{S_i}(g_i, g_{i+1}) \leq \alpha_{i+1}$. Observe that the points in $S_i$ (without their labels) are chosen i.i.d. from $\mathcal{D}$, and hence, By Theorem A.2 (VC bounds) and $|S_i| \geq 128\lambda_i \geq \lambda_{i+1}$, with probability at least $1 - \beta_{i+1}$ we have that $\text{error}_{\mathcal{D}}(g_i, g_{i+1}) \leq \alpha_{i+1}$. Hence, with probability $1 - 2\beta_{i+1}$, we have $\text{error}_{\mathcal{D}}(g_i, g_{i+1}) \leq \alpha_{i+1}$. Finally, by the triangle inequality, $\text{error}_{\mathcal{D}}(g_{i+1}, c) \leq \sum_{j=1}^{i+1} \alpha_j$, except with probability $2\sum_{j=1}^{i+1} \beta_j$ $\square$

Define the following good event.

> **Event $E_1$:**   Algorithm `GenericBBL` never fails on the execution of `BetweenThresholds` in step 3(d)iv.

**Claim E.2.** *Event $E_1$ occurs with probability at least $1 - \beta$.*

*Proof.* Using to union bound and Claim D.5,

$$\Pr[\text{Event } E_1 \text{ occurs}] \geq 1 - \beta.$$

$\square$

Combining claims E.1 and E.2, we get:

**Claim E.3.** *Let $\mathcal{D}$ be an underlying distribution and let $c \in C$ be a target concept. Then*

$$\Pr[\forall i \; \exists g_i \in C \text{ s.t. } \text{error}_{S_i}(g_i) = 0 \text{ and } \text{error}_{\mathcal{D}}(g_i, c) \leq \alpha] \geq 1 - 3\beta.$$

**Notations.** Consider the $i$th phase of Algorithm `GenericBBL`, and focus on the $j$-th iteration of Step 3. Fix all of the randomness in `BetweenThresholds`. Now observe that the output on step 3(d)iii is a deterministic function of the input $x_{i,j}$. This defines a hypothesis which we denote as $h_{i,j}$.

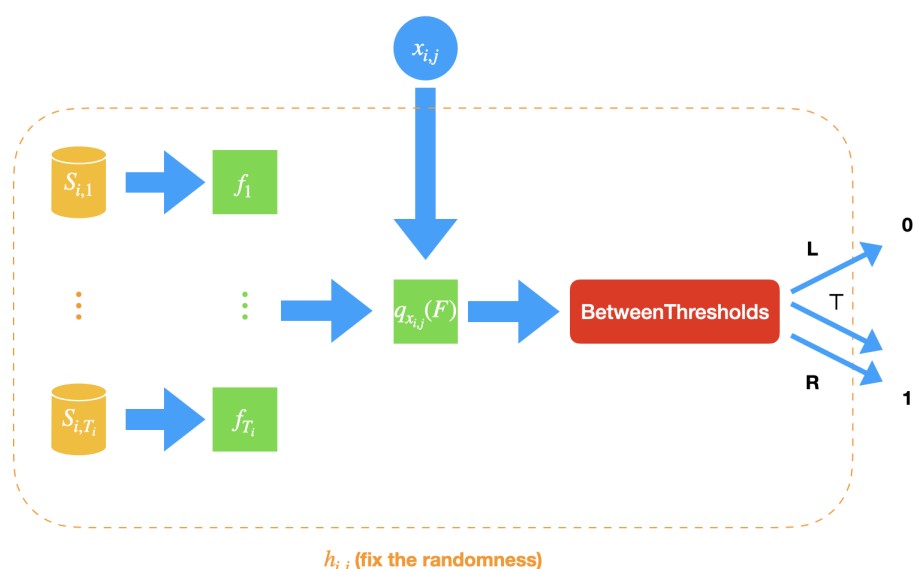

Figure 3: Hypothesis $h_{i,j}$

**Claim E.4.** *For $\beta < 1/16$, with probability at least $1 - 4\beta$, all of the hypotheses defined above are $6\alpha$-good w.r.t. $\mathcal{D}$ and $c$.*

*Proof.* In the phase $i$, by Claim E.3, with probability at least $1 - 3\beta$ we have that $S_i$ is labeled by a hypothesis $g_i \in C$ satisfying $\mathrm{error}_{\mathcal{D}}(g_i, c) \le \alpha$. We continue with the analysis assuming that this is the case.

On step 3a of the $i$th phase we divide $S_i$ into $T_i$ subsamples of size $\lambda_i$ each, identify a consistent hypothesis $f_t \in C$ for every subsample $S_{i,t}$, and denote $F_i = (f_1, \ldots, f_T)$. By Theorem A.2 (VC bounds), every hypothesis in $F_i$ satisfies $\mathrm{error}_{\mathcal{D}}(f_t, g_i) \le \alpha$ with probability $3/4$, in which case, by the triangle inequality we have that $\mathrm{error}_{\mathcal{D}}(f_t, c) \le 2\alpha$.

Set $T_i \ge \frac{512(1-4\beta_i)\ln(\frac{1}{\beta_i})}{(1-64\beta_i)^2}$, using Chernoff bound, it holds that for at least $15T_i/16$ of the hypotheses in $F_i$ have error $\mathrm{error}_{\mathcal{D}}(f_t, g_i) \le 2\alpha$ with probability at least $1 - \beta_i$. These hypotheses have $\mathrm{error}_{\mathcal{D}}(f_t, c) \le 3\alpha$.

Let $m : X \to \{0, 1\}$ defined as $m(x) = \mathrm{maj}_{f_t \in F_i}(f_t(x))$. For $m$ to err on a point $x$ (w.r.t. the target concept $c$), it must be that at least $7/16$-fraction of the $3\alpha$-good hypotheses in $\hat{F}_i$ err on $x$. Consider the worst case in Figure 4 , we have $\mathrm{error}_{\mathcal{D}}(m, c) \le 6\alpha$

By Lemma C.4 and Claim D.7, with probability at least $1 - \beta_i$, all of the hypotheses defined during the $i$th iteration satisfy this condition, and are hence $6\alpha$-good w.r.t. $c$ and $\mathcal{D}$. By the union bound,

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

}_1, \ldots, D^{b,L*}_i, D^{b,L}_{i+1})$, and $(D^{b,L}_1, \ldots, D^{b,L}_{i-1}, D^{b,L*}_i)$ follow the same distribution for $b \in \{0,1\}$,
644 where $D^{b,L*}_i$ is the labels of points in $D^b_i$ expect the different point. So that it suffices to show
645 that $\left(D^{0,L}_{i+1}, S^0_2\right)$ and $\left(D^{1,L}_{i+1}, S^1_2\right)$ are $(\varepsilon,\delta)$-indistinguishable.

We follow the processes creating $D_{i+1}^{b,L}$ and $S_{i+2}^b$ in Figure 6: (i) The mechanism $M_1$ corresponds to the loop in Step 3d of GenericBBL where labels are produced for the adversarially chosen points $D_{i+1}^b$. By application of Lemma C.3, $M_1$ is $(1,\delta)$-differentially private. (ii) The mechanism $M_2$, corresponds to the subsampling of $\hat{S}_{i+1}^b$ from $S_{i+1}^b$ and the application of procedure LabelBoost on the subsample in Step 3f of GenericBBL resulting in $S_{i+2}^b$. By application of Claim 2.7 and Lemma C.1, $M_2$ is $(\varepsilon, 0)$-differentially private. Thus $(M_1, M_2)$ is $(\varepsilon + 1, \delta)$-differentially private. (iii) The mechanism $M_3$ with input of $\hat{D}_i^b$ and output $\left(D_{i+1}^{b,L}, S_{i+2}^b\right)$ applies $(M_2, M_3)$ on $S_{i+1}$, which is generated from $\hat{D}_i^b$ and in Step 3f of GenericBBL. By application of Claim C.1, $M_3$ is $(\varepsilon + 4, 4\varepsilon\delta)$-differentially private. (iv) The mechanism $M_4$, corresponds to the subsampling $\hat{D}_i^b$ from $D_i^b$ and the application of $M_4$ on $\hat{D}_i^b$. By application of Claim 2.7, $M_4$ is $(\varepsilon, \frac{16e\varepsilon\delta}{3+\exp(\varepsilon+4)})$-differentially private. Since $\frac{16e\varepsilon}{3+\exp(\varepsilon+4)} \leq 1$ for any $\varepsilon$, $\left(D_{i+1}^{0,L}, S_2^0\right)$ and $\left(D_{i+1}^{1,L}, S_2^1\right)$ are $(\varepsilon, \delta)$-indistinguishable. $\qquad\square$

**Remark E.7.** *The above proofs work on the adversarially selected D because: (i) Lemma C.3 works on the adaptively selected queries. (We treat the hypothesis class $F_i$ as the database, the unlabelled points $x_{i,\ell}$ as the query parameters.) (ii) LabelBoost generates labels by applying one private hypothesis on points. The labels are differentially private by post-processing.*