# OpenReview forum: "Private Everlasting Prediction"
_NeurIPS.cc/2023/Conference — NeurIPS 2023 oral_

### Official Review · Reviewer_NTTD · 2023-07-04

**Soundness:** 3 good
**Presentation:** 4 excellent
**Contribution:** 3 good
**Rating:** 7
**Confidence:** 3

**Summary:**

This work proposes the notion of private everlasting prediction. Given a training dataset, the predictor responds to a sequence of queries and privacy has to be preserved for both the training data and all queries. The authors explore the PAC learnability problem under this model and show that the sample complexity scales quadratically with the VC dimension through a generic construction from non-private learners.

**Strengths:**

1. The authors formulate a new notion of private everlasting prediction. This is an original theoretical concept which extends the single query prediction model and has practical relevance.
2. The authors prove several interesting theoretical properties for private everlasting prediction. First it requires that the hypothesis needs to change over time. Second, the authors show that the sample complexity scales quadratically with the VC dimension of the concept class. This is a significant improvement compared to private learning which is impossible over infinite domains.
3. The writing is clear and easy to follow.

Update: increased my score to seven after seeing the authors' response and other reviewer comments.

**Weaknesses:**

1. In the algorithm GenericBBL, $\tau>1.1\times 10^10$ seems too large. Can the constant be made any smaller?
2. The algorithm is not computationally efficient.
3. In terms of writing, the authors could provide a brief overview of the proof ideas for Theorem 5.1.

**Questions:**

In practice, some large models are pretrained on public data which can be considered non-private. Theoretically, is it relevant to consider the case when the training set $S$ is non-private and the stream of queries is private? How would the sample complexity change?

**Limitations:**

The authors adequately addressed the limitations and potential negative societal impact.

---

> ### Author Rebuttal · Authors · 2023-08-08
>
> Thank you for your thoughtful comments. Below we address the points you make:
>
> **> In the algorithm GenericBBL,  $\tau>1.1\times 10^{10}$, seems too large. Can the constant be made any smaller?**
>
> We have not optimized constants as our contribution focuses on asymptotic complexity. The  paper introduces a new concept - private everlasting prediction - and demonstrates an unbounded asymptotic improvement over private learning. While the construction may be impractical at this stage, we believe that it provides a significant theoretical improvement that would be followed by more theoretical and practical research towards making everlasting predictors usable in a variety of practical applications.
>
> **> The algorithm is not computationally efficient**
>
>  Although our algorithm is not computationally efficient it does yield in some cases polynomial time constructions. We clearly state the question of making the construction efficient as an open problem. We expect that future work would resolve the question whether an efficient generic construction exists.
>
> **> In terms of writing, the authors could provide a brief overview of the proof ideas for Theorem 5.1.**
>
> We will include an overview of the proof. Very briefly - the high-level idea is to use LabelBoost to label the unlabeled dataset and use BetweenThreshold to predict labels. The privacy and accuracy guarantees come from these two algorithms.
>
> **> In practice, some large models are pretrained on public data which can be considered non-private. Theoretically, is it relevant to consider the case when the training set S is non-private and the stream of queries is private? How would the sample complexity change?**
>
> Variations of this question were studied in the standard private learning model (where the learner releases a model), e.g., by [1,2]. We believe that this could also be relevant in our context, but this would require adjustment to the learning scenario to be meaningful. More specifically, in our context, if pre-training can be performed on O(VC) non-private labeled examples, then that would nullify the need for private learning/prediction, as we could simply release the resulting non-private model.
>
> We believe that this would make sense in other learning scenarios where you would like to use the private queries in order to improve the error of the non-private model you obtained from the initial (non-private) training set. Our privacy definition could potentially fit such scenarios.
>
> [1] Beimel, Nissim, Stemmer. Private Learning and Sanitization: Pure vs. Approximate Differential Privacy. RANDOM 2013.\
> [2] Bassily, Moran, Alon. Limits of Private Learning with Access to Public Data. NeurIPS 2019

---

> > ### Comment · Reviewer_NTTD · 2023-08-14
> > **Thank you for your resonse**
> >
> > Thank you for your response. I believe the new concept is very meaningful and could inspire a new line of research. I will increase my score to 7.

---

### Official Review · Reviewer_nCbQ · 2023-07-04

**Soundness:** 3 good
**Presentation:** 2 fair
**Contribution:** 2 fair
**Rating:** 7
**Confidence:** 3

**Summary:**

The paper discusses private everlasting prediction, which extends private prediction to answer an unlimited sequence of prediction queries. The goal is to present a generic private everlasting predictor with low training sample complexity. The paper introduces definitions for everlasting prediction and everlasting differentially private prediction interfaces. It then presents a generic construction called GenericBBL for private everlasting prediction.

**Strengths:**

1. This paper introduces a formal framework for everlasting prediction.
2. This paper provided a comprehensive privacy analysis to the proposed privacy-preserving everlasting prediction task.

**Weaknesses:**

1. Limited applicability: The paper focuses on the theoretical aspects of private everlasting prediction and does not provide concrete practical applications or empirical evaluations. It remains to be seen how well the proposed approach translates into real-world scenarios.
2. Lack of comparative experimental analysis: The excerpt does not mention any comparison or benchmarking against existing methods or alternative approaches. Without such comparisons, it is difficult to assess the novelty or superiority of the proposed method.
3. Lack of empirical validation: The excerpt does not mention any empirical experiments, simulations, or case studies to validate the effectiveness or robustness of the proposed private everlasting predictor. It is unclear how the proposed construction performs in terms of privacy preservation, utility, and sample complexity compared to other techniques.
4. The contents shown in the supplementary materials contain a lot of redundant information compared with the paper.

**Questions:**

What is the application scenario of the everlasting prediction?

**Limitations:**

It lacks experimental analysis of the proposed algorithm, which makes it hard to justify the utility of the proposed algorithm.

---

> ### Author Rebuttal · Authors · 2023-08-08
>
> Thank you for your thoughtful comments. Below we address the points you make:
>
> **> Limited applicability: The paper focuses on the theoretical aspects of private everlasting prediction and does not provide concrete practical applications or empirical evaluations**
>
> We agree that the construction of efficient practical algorithms that can be used in concrete practical applications is important. Establishing a sound theory of private learning and improving the asymptotic behavior of private learning algorithms is no less important. In fact many if not most of the practical applications of differential privacy would not have existed hadn't they began as concepts which were introduced in theoretical results, even if initial constructions were impractical.
>
> **> Lack of comparative experimental analysis**
>
> We present a novel concept (everlasting prediction) as an alternative to private learning. We also present a construction demonstrating that everlasting prediction is possible in tasks where private learning was proved to be impossible. For example, the sample complexity for private learning of threshold functions grows with the domain size. In contrast, as we mention in the paper, threshold functions can be predicted efficiently, regardless of the domain size.
>
> **> Lack of empirical validation... makes it hard to justify the utility of the proposed algorithm**
>
> At this phase of this research, an experimental analysis and an empirical validation would not yield meaningful conclusions beyond deeming the current algorithms "impractical". Rather, a mathematical asymptotic analysis is the right tool for analyzing our results. We believe that our work would lead to future studies on this topic, both theoretically and practically oriented.
>
> **> The contents shown in the supplementary materials contain a lot of redundant information compared with the paper**
>
> The supplementary materials contain all the formal details which were omitted from the main paper. We are open to reorganizing the paper (within the page limit).
>
> **> What is the application scenario of the everlasting prediction?**
>
> Essentially, private everlasting predictors can be used in many scenarios where we would like to use the outcome of private learning algorithms for prediction, e.g.,
>
> 1. A hospital might use a private everlasting predictor in supporting decisions whether patients need to be treated for COVID19, based on their tests and medical history. The process would bootstrap with an initial sample of patients that would be labeled by experts and continue with private prediction. Due to the very sensitive nature of the data (as well as legal and ethical considerations) it is important to protect the information of both the initial sample and of patients to whom prediction is applied.
>
> 2. Similarly, a bank might use a private everlasting prediction as part of its decision process whether to offer loans to customers.

---

> > ### Comment · Reviewer_nCbQ · 2023-08-15
> > **Thank you for your response**
> >
> > Thank you for the explanation! It addresses my concerns and I agree that the new concept is very meaningful. I would like to increase my score to a 7.

---

### Official Review · Reviewer_xv4Q · 2023-07-07

**Soundness:** 4 excellent
**Presentation:** 4 excellent
**Contribution:** 4 excellent
**Rating:** 8
**Confidence:** 3

**Summary:**

This paper provides an intriguing path to evading known lower bounds for differentially private PAC learning. Whereas nonprivately the sample complexity of learning is proportional to the VC dimension, the private sample complexity for (pure) DP is characterized by the representation dimension, which can be much larger. In particular, some natural classes such as threshold functions over an infinite domain (e.g. Z or R) have finite VC dimension but infinite representation dimension, so they can be learned nonprivately but not privately (even with approximate DP, by a separate result).

This paper shows that this impossibility only holds for privately releasing a hypothesis and not for privately classifying samples. In particular, it shows that in the online prediction setting, for any hypothesis class there is a generic stateful algorithm with sample complexity based on the VC dimension (to be precise, quadratic in the VC dimension) that can privately answer an unbounded sequence of iid queries with the same accuracy guarantee as PAC learning, but where the state of the algorithm remains hidden and only the query labels are revealed. The algorithm is allowed to remember previous queries but must be differentially private with respect to them as well as the points of the original (labeled) training set. It also shows that statefulness is necessary to achieve this result.

For general hypothesis classes the algorithm is inefficient, but it can be made efficient on important special cases, including threshold functions.

**Strengths:**

This is a very interesting result that enhances our understanding of differentially private learning by bypassing known lower bounds. It essentially gives a separation between interactive and non-interactive private prediction, showing that interactivity reduces the private sample complexity to nearly that of nonprivate learning (as long as we only have to output predictions and not a hypothesis). The paper is quite well-written.

**Weaknesses:**

There are two obvious limitations of the main result, that yield very interesting open problems posed in the paper: whether it's possible to come up with a similar algorithm that is computationally efficient for all hypothesis classes, and whether it's possible to improve the sample complexity from VC^2 to VC.

Minor corrections:
104: points --> point
147: "upto" should be "up to"

**Questions:**

1) Is it strictly necessary for accuracy that the query points are iid? Or could we hope for the following stronger statement:

Given query points x_1, x_2, ... and a subset of indices I (not known to the algorithm) such that x_i is sampled from the distribution for i\in I but is chosen by the adversary for i\notin I, we require accuracy only for the points in I.

That is, as long as the challenge queries are sampled from the distribution, we want to be robust to data poisoning using the other queries. Does this seem like it could be possible, or is it too strong?

2) Minor clarification: In the technical overview on page 3-4 it sounds like the parameters discussed are based on the standard composition bound, without using BetweenThresholds. It might be helpful to clarify whether this is the case and if we get a similar or only slightly weaker result without using BetweenThreshold, or if we truly need the tighter privacy analysis of BetweenThreshold.

**Limitations:**

Yes.

---

> ### Author Rebuttal · Authors · 2023-08-08
>
> Thank you for your thoughtful comments. Below we address the points you make:
>
> **> There are two obvious limitations of the main result that yield very interesting open problems… computational efficiency and sample complexity**
>
> We believe that - even with these questions unresolved - the concept of everlasting prediction is of interest to the private learning community, in particular because it gets around impossibility results. We hope that follow up work will resolve the open questions we present in the paper.
>
> **> Is it strictly necessary for accuracy that the query points are iid? ... Does this seem like it could be possible, or is it too strong?**
>
> We believe that a construction similar to ours should work in a setting where not too many of the challenge queries are chosen adversarially, provided that the given non-private learners can withstand a poisoning attack with related parameters. We view this research direction (and in particular the stronger formulation you suggested) as an interesting direction for future work.
>
> **> Minor clarification ... can we get a similar or only slightly weaker result without using BetweenThreshold**
>
> We can get a similar (but weaker) result even without BetweenThresholds. Specifically, this will yield an algorithm in which the initial labeled sample S has a worse dependency on the accuracy parameter alpha.

---

### Official Review · Reviewer_9JKU · 2023-07-07

**Soundness:** 4 excellent
**Presentation:** 3 good
**Contribution:** 4 excellent
**Rating:** 9
**Confidence:** 4

**Summary:**

This work studies differentially private prediction. It has two major contributions:

a) Prediction corresponds to being given an initial labeled training set, and then subsequently making predictions on other data points based on it. This paper shows that differentially private prediction can be performed on an unbounded number of queries, with strong accuracy guarantees, with the initial training set sample complexity scaling only as a polynomial in the VC dimension of the hypothesis class. Prior work either studied prediction for a small number of queries or studied the more stringent task of private PAC learning where significantly stronger lower bounds are known (PAC learning asks for the release of an entire model as opposed to predictions alone).

b) Since queries correspond to data points, they are usually sensitive user information. The authors formalize a model of privacy (similar to joint differential privacy) that gives a meaningful notion of privacy for these data points, even when adversaries can choose the queries adaptively. Their algorithms satisfy this notion of privacy.


**Strengths:**

This paper makes a valuable and somewhat surprising discovery by proving that differentially private prediction can be performed on an unbounded number of queries, with no training set sample complexity dependence on the number of queries! They leverage standard techniques such as sparse vector in clever ways to do so. They operate in rounds, and use queries themselves as data points for future rounds, calling upon techniques from the semi-supervised differentially private learning literature . Their results suggest new ways to get around lower bounds for differential privacy.

Their algorithms involve reducing private prediction to non-private PAC learning, and while their techniques are not yet practical in many cases (because of time complexity), their reduction can be readily extended to any non-private learning algorithm, and hence there is lots of potential for future work to come up with ways to make their techniques more practical.


**Weaknesses:**

One (minor) drawback is that the notion of privacy defined for the adaptively chosen queries is (necessarily) not as strong as would be nice since the label for a data point needs to depend on the data point to achieve any reasonable notion of accuracy (since otherwise you can only do something like randomized response).

**Questions:**

1) The reasoning for the use of LabelBoost didnt entirely make sense to me- the stated reason was that the predictions provided during a round are not consistent with any concept in the hypothesis class. However, it’s not immediately clear to me that this is a problem- for example, using agnostic (non-private) learning algorithms may get around this issue (at the expense of worse sample complexity dependence on the accuracy parameter). More explanation of this would be useful.

2) The comment about their model of prediction, in principle, allowing memorization was interesting. It’s not clear that this is true- it seems like memorizing something in the training set would affect future predictions which would affect the adversary’s views. Would love more clarification on this.

3) Is there a simple (eps, 0)- version of this algorithm? It’s not clear to me that approx DP is necessary (sparse vector by itself is pure DP, though a variant is used in this paper).

4) The diagrams in the privacy proofs are a little too convoluted to be useful to the reader. I don’t have great suggestions on how to simplify them, but the privacy proofs by themselves are relatively simple, and I’m not sure the diagrams make them simpler to understand.

5) Some minor comments:

a) In Algorithm LabelBoost step 4, ‘choose’ based on exponential mechanism is not well defined since the score function is not specified (it can be inferred from context that it’s the negative of empirical risk) but specifying would be good.

b) In Claim E.4 don’t you need to account for the fact that the function labeling the sample is not the real function, but rather the one chosen by LabelBoost? Specifically, Claim E.3 is about a function with 0 sample error, whereas LabelBoost will incur some error, and hence I think you want a version of claim E.3 where sample error is bounded by alpha.

c) On page 8, where it says ‘using standard privacy amplification arguments, BetweenThresholds can be modified to allow for c times of outputting bot before halting)’, I think you mean privacy composition.

**Limitations:**

Limitations adequately addressed.

---

> ### Author Rebuttal · Authors · 2023-08-08
>
> Thank you for your thoughtful comments. Below we address the points you make:
>
> **> the notion of privacy defined... is not as strong as would be nice since the label for a data point needs to depend on the data point**
>
> We don’t see the fact that the adversary does not get to see the label of one point (where the inputs differ) as a serious weakness - this is a necessity due to the characteristics of the problem, and it makes sense in may applications of private prediction (e.g., when prediction is used to support a decision whether to give a patient a certain treatment, it is OK for the patient to learn the result of applying the prediction to their own medical information; what is important is that privacy of other patients needs to be preserved). Note that the adversary we consider is nevertheless very powerful. In particular, after the single point where it does not see the label it can still adaptively choose an unbounded number of points and learn their labels.
>
> **> The reasoning for the use of LabelBoost didnt entirely make sense to me**
>
> In our construction we need to ensure that the sum of $\alpha_i$ converges, so that the overall error is under control. LabelBoost helps here because its utility guarantees are better than what we can guarantee during "runtime", i.e., when responding to the queries online. We do not know if LabelBoost is necessary for this, but so far we could not get the error to converge without it.
>
> **> The comment about their model of prediction, in principle, allowing memorization was interesting. It's not clear that this is true- it seems like memorizing something in the training set would affect future predictions which would affect the adversary's views**
>
> Memorization need not be of just the initial training set but also of the query points posed to the predictor. What we meant to say is that our construction yields private predictions regardless of how the internal non-private learners operate, and even in case they memorize points of the training set or points presented as queries.
>
> A possible example for this would be the class $C^{enc}_{thresh}$ (described in Item 1 in the same section). Bun and Zhandry [2016] showed that privately learning this class is computationally hard, whereas non-private learning is easy. This difference actually follows from the private learner's inability to memorize data points from the training set. As we mentioned in Item 1, our construction yields an efficient learner for this class (in our privacy model). The resulting learner would not leak to the adversary information about data points, but it would still memorize some of them internally. We will include this observation in the final version.
>
> **> Is there a simple (eps, 0)- version of this algorithm?**
> We don't know whether pure-DP everlasting predictors exist. Our analysis strongly relies on the advanced composition theorem, and hence yields approximate-DP.
>
> **> The diagrams in the privacy proofs are a little too convoluted to be useful to the reader**
> Thank you for the comment. We will consider whether to keep the diagrams (or simplify them).
>
> **> In Algorithm LabelBoost step 4, 'choose' based on exponential mechanism is not well defined (it can be inferred from context)**
>
> Right. We will state this explicitly.
>
> **> In Claim E.4 don't you need to account for the fact that the function labeling the sample is not the real function, but rather the one chosen by LabelBoost?**
>
> Yes, the samples are labeled by the concept chosen by LabelBoost. We argue that this concept has low error compared with the original concept in claim E.3
>
> **> On page 8... I think you mean privacy composition**
>
> Yes, we use advanced composition because BetweenThreshold will halt within c times with high probability.

---

> > ### Comment · Reviewer_9JKU · 2023-08-18
> >
> > Thanks for your responses!

---

### Decision · Program_Chairs · 2023-09-21

**Decision:**

Accept (oral)

**Comment:**

The paper got a very strong set of reviews. The reviewers, and I will recommend the authors to update the presentation in light of the discussions.